# A Comparison of Monocular Visual SLAM and Visual Odometry Methods Applied to 3D Reconstruction

Erick P. Herrera-Granda [1,2,3,*] (ID), Juan C. Torres-Cantero [2] (ID), Andrés Rosales [4,5]
and Diego H. Peluffo-Ordóñez [3,6,7] (ID)

1  Unidad de Educación en Línea, Universidad de Otavalo, Otavalo 100202, Ecuador
2  Department of Computer Languages and Systems, University of Granada, 18071 Granada, Spain; jctorres@ugr.es
3  SDAS Research Group, Ben Guerir 43150, Morocco; peluffo.diego@um6p.ma
4  GIECAR, Departamento de Automatizacióny Control Industrial, Escuela Politécnica Nacional, Quito 170525, Ecuador; andres.rosales@epn.edu.ec
5  Universidad de Investigación de Tecnología Experimental Yachay, San Miguel de Urcuquí 100115, Ecuador
6  College of Computing, Mohammed VI Polytechnic University, Salé 43150, Morocco
7  Faculty of Engineering, Corporación Universitaria Autónoma de Nariño, Pasto 520001, Colombia
*  Correspondence: erickherreraresearch@gmail.com; Tel.: +593-989460084

**Abstract:** Pure monocular 3D reconstruction is a complex problem that has attracted the research community's interest due to the affordability and availability of RGB sensors. SLAM, VO, and SFM are disciplines formulated to solve the 3D reconstruction problem and estimate the camera's ego-motion; so, many methods have been proposed. However, most of these methods have not been evaluated on large datasets and under various motion patterns, have not been tested under the same metrics, and most of them have not been evaluated following a taxonomy, making their comparison and selection difficult. In this research, we performed a comparison of ten publicly available SLAM and VO methods following a taxonomy, including one method for each category of the primary taxonomy, three machine-learning-based methods, and two updates of the best methods to identify the advantages and limitations of each category of the taxonomy and test whether the addition of machine learning or updates on those methods improved them significantly. Thus, we evaluated each algorithm using the TUM-Mono dataset and benchmark, and we performed an inferential statistical analysis to identify the significant differences through its metrics. The results determined that the sparse-direct methods significantly outperformed the rest of the taxonomy, and fusing them with machine learning techniques significantly enhanced the geometric-based methods' performance from different perspectives.

**Keywords:** monocular 3D reconstruction; monocular SLAM comparison; monocular VO comparison; monocular benchmark; 3D reconstruction classification





## 1. Introduction

Monocular 3D reconstruction is a complex problem that can be solved from multiple perspectives (commonly requiring combining geometric, probabilistic, and even machine learning techniques), due to the large amount of information to be processed and the scale ambiguity problems that pure monocular sensors imply [1,2]. This problem has been studied in the past three decades to obtain 3D representations of an environment using a sequence of images as the unique source of information for an algorithm. Previously, multiple researchers have explored the possibility of addressing this problem by using diverse hardware like radars, lasers, GPS, INS, cameras, and any possible combination thereof. Regarding the camera alternative, it can be combined with active or passive infrared sensors as RGB-D input modalities. It can also be structured as an array of cameras registering the same objects from multiple angles to allow triangulation. Monocular RGB

sensors can also be used alone to register a frame sequence from which the algorithm can process a scene from multiple views [3,4]. This last option is known as monocular RBG or monocular pure visual input modality, used in monocular Simultaneous Landing and Mapping (SLAM), Visual Odometry (VO), or Structure from Motion (SFM) to obtain 3D reconstructions of environments and estimate the ego-motion of an agent from such representations. In recent years, the pure monocular input modality has attracted the research community's attention due to the sensors' low price and availability in most handheld devices—smartphones, tablets, and laptops. Thus, monocular SLAM, VO, and SFM systems are not limited as other sensors are (like lasers or radars) to work in a limited range and have demonstrated the ability to recover precise trajectories and 3D reconstructions indoors and outdoors.

Simultaneous Localization and Mapping is the process where a robot constructs a map of its surroundings while concurrently figuring out where it is located within that map. It involves determining the positions of landmarks and objects near the robot and its position, commonly utilizing sensors and geometric and Bayesian techniques. Visual Odometry is the process of incrementally estimating the robot's ego-motion (location and orientation) by analyzing the changes between the sequential camera images from the robot, estimating the robot's local trajectory rather than obtaining a comprehensive map. VO is commonly utilized as a front-end in many visual SLAM systems. Structure from Motion refers to a reduction in the 3D structure from 2D image sequences that show a scene from different perspectives. It recovers the 3D location of points matched across multiple images and the camera pose for each image. SFM does not require knowing the camera's motion in advance and is utilized in SLAM for initializing new 3D points [5].

As mentioned before, SLAM, VO, and SFM are three disciplines that can be used to achieve the 3D reconstruction goal. SLAM is a discipline that appeared in the robotics field motivated by the objective of estimating the environment map from where the trajectory of a robot can be calculated, which can be used for autonomous navigation, driving, and flying, among other things. In the computer vision field, multiple systems have been created to address similar problems: SFM and VO. Structure from Motion specializes in recovering an environment geometry, while Visual Odometry focuses on calculating the trajectory and pose of a moving camera. However, it has been demonstrated that instead of solving each problem separately, the best results have always been obtained by solving and optimizing both problems simultaneously [3,6–8]. That is why it is common to find VO methods that include SFM modules to improve their performance and SFM methods that use VO to improve estimation or optimization tasks. For such reasons, in this study, we aim to identify the best monocular RGB methods for 3D reconstruction; so, we included methods from these three disciplines suitable for recovering 3D environment reconstructions.

As a complex problem, pure visual monocular 3D reconstruction has been addressed from multiple perspectives combining various techniques that can be classified following different approaches. One early classification is described in the study of [3] defining the feature-based and appearance-based categories; nevertheless, this approach is unsuitable for covering all the SLAM, VO, and SFM techniques available nowadays in the state of the art. A better approach to classify monocular RGB 3D reconstruction systems is the taxonomy described in [9], considering three classifications covering dense, sparse, direct, indirect, classic, and machine learning-based proposals. Moreover, the authors listed 42 methods classified following the proposed extended taxonomy. After a careful reading and analysis of the 42 listed methods, we could identify that many of the existing methods were not adequately evaluated on large datasets [10–13] or not tested under different motion patterns and illumination changes [11,14,15] and not tested for indoors/outdoors [16–18]; or the results were not obtained on the same metrics [8,19,20] hindering comparison and selection. In addition, most of the methods performed comparisons against the currently available methods from the state of the art, providing results in tables summarizing the average mean or median of the algorithm execution on a specific scene, but they did not provide an inferential statistical analysis of the results; thus, the reported differences or

improvements cannot be considered significant. Moreover, given the fact that before the extended taxonomy described in [9], there were only general classifications like direct vs. indirect and sparse vs. dense methods [7,8,21] or the feature-based and appearance-based classification reported in [3], none of the studies compared their results following a taxonomy that might allow identifying better the advantages and limitations of direct, indirect, dense, and sparse methods.

To address the mentioned issues, in this study, we performed a comparison of ten publicly available SLAM and VO methods following a taxonomy, where the main contributions are:

- A comparison of 10 SLAM and VO methods, following the main classification described in the taxonomy (sparse-indirect, dense-indirect, dense-direct, and sparse-direct), to identify the advantages and limitations of each method of those classifications.
- A comparison of three machine learning-based methods against their classic geometric versions to identify whether there are significant improvements in adding neuronal networks to classic approaches.
- An inferential statistical analysis describing the procedure to identify significant differences based on the most suitable metrics for testing monocular RGB methods.

We also provide video samples of each algorithm's execution as Supplementary material in the GitHub repository: "https://github.com/erickherreraresearch/MonocularPureVisualSLAMComparison accessed on 16 June 2023", along with all the .txt result files of each algorithm run for reproducibility.

### 1.1. Related Works

Following the classification described in [21], there are four main classifications for the methods that can be used to recover a scene's 3D geometry using monocular image sequences as the unique source of information: sparse-indirect, dense-indirect, dense-direct, and sparse-direct.

### 1.1.1. Sparse-Indirect Methods

Sparse-indirect methods implement preprocessing steps recovering sparse reconstructions. MonoSLAM, PTAM, ORB-SLAM, and OpenMVG are the most prominent works in this classification. MonoSLAM [22] was one of the first real-time monocular SLAM systems. Its key contributions included using large image patches as features, "active" feature matching based on uncertainty, and initializing by tracking known targets. However, MonoSLAM was limited to small workspaces and lacked loop-closing abilities. PTAM [23] introduced the concept of parallel tracking and mapping threads, with the map optimized via bundle adjustment over carefully selected keyframes. This configuration achieved excellent AR tracking in small spaces, but the PTAM lacked loop closing, and the relocalization was view-dependent. ORB-SLAM [24] significantly expanded PTAM's capabilities using ORB features for tracking, mapping, and loop closing via DBoW2 place recognition. The covisibility graphs enabled local mapping, while the pose graphs distributed loop closures globally. ORB-SLAM also introduced flexible keyframe insertion/deletion policies to improve mapping during exploration while reducing redundancy. This versatility enabled state-of-the-art performance across indoor, outdoor, handheld, and robotics datasets. OpenMVG is a C++ library that provides an interface to multiple view geometry algorithms for building complete 3D reconstruction pipelines from images implementing incremental and global SfM approaches. The OpenMVG SfM pipeline stores camera poses, landmarks, and observations, providing smooth data flow between OpenMVG modules. Overall, the OpenMVG enables flexible experimentation and the development of new techniques used for multiple implementations since 2016; however, it only allows recovering widely sparse reconstructions, which are unsuitable for many applications.

### 1.1.2. Dense-Indirect Methods

Dense-indirect techniques incorporate preprocessing stages and recover dense depth maps. Some important prior works that defined this category were Valgaerts et al. and Ranftl et al. Valgaerts et al. [25] proposed a novel two-step method for estimating the fundamental matrix from a dense optical flow. Their key contribution was demonstrating that accurate epipolar geometry robust estimation was possible using dense correspondence fields computed by variational optical flow methods. They introduced a joint variational model that recovered the optical flow and epipolar geometry within a single energy functional, thus improving the results. However, their method was limited by its sensitivity to large displacements and occlusions. Ranftl et al. [26] presented an approach to estimate dense depth maps for complex dynamic scenes from monocular video, built on the use of dense optical flow. The key concept is a motion segmentation stage that decomposes the scene into independent rigid motions, each with its epipolar geometry enabling moving objects' reconstruction. Its method was optimized to work with object scales and geometry to assemble a globally consistent 3D model determined up to scale. A key difference from Valgaerts et al. was the explicit handling of multiple independently moving objects and the recovery of dense depth for fully dynamic scenes. However, Ranftl et al.'s approach still relied on approximate scene rigidity and the connectivity of objects to the environment. Valgaerts et al. introduced a dense optical flow for fundamental matrix estimation, while Ranftl et al. extended dense the geometric reconstruction to complex dynamic scenes. Both moved from sparse features to dense correspondence fields; in contrast, Ranftl et al. focused on depth estimation and scene assembly.

### 1.1.3. Dense-Direct Methods

Dense-direct techniques work directly with pixel information and can recover dense depth maps. Some of the main contributions in this field are the Stühmer et al., DTAM, REMODE, and LSD-SLAM systems. Stühmer et al. [27] proposed one of the first real-time dense monocular SLAM systems. They introduced a variational framework to estimate the dense depth maps from multiple images using robust penalizers for both the data term and the regularizer. The key contributions were integrating multiple images for noise robustness and an efficient primal-dual optimization scheme. However, their method was limited to local dense tracking and mapping without global map optimization. The DTAM system proposed by Newcombe et al. [26] enabled real-time dense tracking and global mapping using a single handheld camera. They introduced the concept of dense model-based camera tracking by aligning live images to the textured 3D surface models synthesized from the estimated dense depth maps. The depth maps were computed by filtering over the small-baseline stereo comparisons from video. A key difference from Stühmer et al. was maintaining a global map with pose graph optimization. The REMODE system of Pizzoli et al. [28] also performed per-pixel Bayesian depth estimation but introduced a convex optimization-based smoothing step using the estimated uncertainty to enforce the spatial regularity. They demonstrated probabilistic updating, allowing online refinement and error detection. A key contribution was the derivation of a measurement uncertainty model. However, REMODE was limited to local mapping without global optimization. The LSD-SLAM of Engel et al. [29], integrated many of these concepts into the first direct monocular SLAM system capable of performing consistent global semi-dense reconstruction. The key novelties were the direct alignment on the $Sim(3)$ handling scale drift and the incorporation of depth uncertainty into tracking. LSD-SLAM reached an outstanding outdoor performance by enabling large-scale accurate monocular dense reconstruction in real time. In summary, early works, like Stühmer et al. and DTAM, introduced key concepts like multiple image integration, probabilistic depth estimation, and variational optimization, while later methods, like LSD-SLAM, were built on these concepts to enable globally consistent mapping and reconstruction, with fully direct approaches finally demonstrating accurate monocular dense SLAM at scale.



### 1.1.4. Sparse-Direct Methods

Sparse-direct techniques work directly on pixel information but do not use all the pixels, producing sparser maps using fewer computational resources. The main contributions from this classification are the DSO, LDSO, and DSM. Direct Sparse Odometry (DSO) was introduced by Engel et al. [21] as the first direct-sparse VO technique. The DSO operates directly on image intensities, optimizing the photometric error instead of the geometric reprojection error. It represents the geometry using inverse depth parametrization and jointly optimizes all the model parameters in real time using a sliding keyframe window. The DSO demonstrated superior accuracy and robustness compared to indirect methods by utilizing edges and intensity variations in featureless areas. However, as a pure visual odometry technique, the DSO suffers from drift over long trajectories as it marginalizes old points and keyframes. Gao et al. presented the LDSO [30], extending the DSO to a more robust VO system by adding loop closure detection and pose graph optimization. The LDSO adapts the DSO's point selection to favor repeatable corner features and computes the ORB descriptors detecting the loop closures using DBoW2. It then estimates the $Sim(3)$ constraints by minimizing the 2D and 3D errors fusing them with the covisibility graph from DSO's sliding window optimization in a pose graph. While reducing the accumulated drift, the LDSO still lacks a persistent map ignoring the existing information after loop closures. Zubizarreta et al. introduced Direct Sparse Mapping (DSM) [31], the first direct sparse monocular SLAM system with a persistent map enabling point reobservations. The DSM selects active keyframes based on temporal and covisibility constraints using the Local Map Covisibility Window applying a coarse-to-fine optimization scheme and a robust cost function based on the t-distribution to handle challenges in converging when incorporating distant keyframes. The DSM demonstrated increased accuracy in trajectory and mapping on EuRoC compared to the DSO, LDSO, and ORB-SLAM. The ability to reuse existing map points resulted in more consistent maps without duplicates. In brief, the DSO pioneered direct-sparse SLAM and achieved superior odometry compared to the indirect methods. The LDSO extended it to full SLAM by adding loop closure detection and correction to reduce drift, while the DSM took a further step creating the first direct technique with a persistent map, enabling beneficial point reobservations through key innovations in window selection, optimization, and robustification.

### 1.1.5. Machine-Learning-Based Approaches

Recently, a new category emerged, adding machine learning modules to the SLAM, VO, and SFM pipelines. Some of the most prominent approaches are DynaSLAM, SVR-Net, VOLDOR, DROID-SLAM, SDF-SLAM, CNN-SLAM, CodeSLAM, DeepFactors, MonoRec, and CNN-SVO. CNN-SLAM [10] was one of the first systems to incorporate CNN-predicted depth maps into monocular SLAM, overcoming the scale ambiguity issues. It also performed joint semantic segmentation and 3D reconstruction, pioneering multitask learning. DynaSLAM [32] was one of the first attempts to detect and remove dynamic objects from the mapping process using a CNN for segmentation and a multiview geometry approach enabling more robust tracking and mapping in dynamic environments. CodeSLAM [33] incorporated an encoder–decoder CNN for scene geometry into a compact latent code conditioned on image intensities retaining only nonredundant information for joint geometry and motion optimization. The CNN-SVO [11] incorporated CNN depth predictions to initialize the depth filters in SVO, reducing uncertainty and improving mapping. Deep-Factors [2] was built over the basis of CodeSLAM to formulate dense monocular SLAM as a factor graph optimization combining the learned depth priors, the reprojection error, and the photometric error for robust performance. VOLDOR [34] integrated a CNN into its visual odometry pipeline using log-logistic depth residuals and probabilistic inference, eliminating the need for feature extraction or RANSAC, enabling real-time performance. The DROID-SLAM [35] integrated a recurrent neural network to iteratively update camera poses and estimate depth maps through differentiable bundle adjustment. MonoRec [14] addressed the alternative to incorporate mask prediction and depth prediction modules

to enable high-quality monocular reconstruction in dynamic scenes. SDF-SLAM [36] combined classic sparse feature extraction with a CNN for dense depth prediction and semantic segmentation enabling semantic 3D reconstruction while retaining real-time performance. SVR-Net [37] integrated a Support Vector Regression network to estimate 3D keypoint locations, enabling robust tracking in challenging conditions using online learning and graph optimization for map refinement. In summary, machine-learning-based methods progressively incorporated deep learning into sparse indirect SLAM systems to improve the robustness and handle the dynamics, achieving dense reconstruction enabling end-to-end learning. The key innovations included using CNNs for segmentation, depth prediction, semantic segmentation, compact scene encoding, and uncertainty modeling.

### 1.1.6. Comparisons

Regarding comparison studies, an early work that accurately compared monocular visual odometry systems was the study of [38], comparing the state-of-the-art methods of that time, DSO, ORB-SLAM, and SVO, on the TUM-Mono benchmark. The authors found that the DSO system, even being a visual odometry system, outperformed the SLAM method and the popular SVO. In that study, the authors also tested the photometric calibration, the motion bias, and the rolling shutter effect, with the available information provided in the TUM-Mono dataset, finding that the photometric calibration improved the performance of the direct methods considerably, and the motion bias effect was more prominent in the indirect method. In contrast, we compared ten methods following a taxonomy, where the three methods tested in [38] were addressed, exploring the same photometric calibration, motion bias, and rolling shutter effects by applying the TUM-Mono dataset. Then in 2020, Mingachev et al. published two comparisons [39,40] testing first the DSO, LDSO, and ORB-SLAM2 and then the ROS-based methods, DSO, LDSO, DynaSLAM, and ORB-SLAM2, on the TUM-Mono and EuRoC benchmarks, where the authors verified the performance of the algorithms implementing an open-source code in their hardware to determine whether there were improvements in the LDSO and DynaSLAM—updates of the original DSO and ORB-SLAM2. The authors found that the updates achieved slight error reductions over their predecessors on both benchmarks, reported as medians of 10 executions of each algorithm in each sequence.

Comparing those studies, we tested ten methods following a taxonomy to test whether the newer versions improved their previous performance and to identify the advantages and disadvantages in the entire taxonomy. We also provided a complete inferential statistical analysis of each method's performance, not only their median values. In addition, we included machine-learning-based versions of the classic methods in our comparison. One of the most recent related works was the study of [41], which explored the state-of-the-art classification and tested visual and visual–inertial algorithms in the ERoC benchmark. In that work, the authors briefly overviewed the existing methods and reviewed the classic classification of direct, feature-based, and RGB-D methods, adding DSO, ORB-SLAM2, and Vins-Mono methods to their comparison. In contrast, this comparison is focused only on monocular RGB methods; so, we followed an appropriate taxonomy for monocular RGB SLAM and VO systems. In addition, we used the TUM-Mono benchmark and its metrics, which is a broader and more complete benchmark.

## 2. Materials and Methods

For this study, we used a taxonomy, algorithms, benchmarks, and metrics suitable for the monocular SLAM and VO problems discussed in the following sections.

### 2.1. Taxonomy

The prior work [6] described a taxonomy based on three classifications in the literature: direct vs. indirect, dense vs. sparse, and classic vs. machine learning.

- **Direct vs. indirect.** Indirect methods refer to those algorithms that implement preprocessing steps, like feature extraction or optical flow estimation, before their pose and

map estimation processes; so, the amount of information that moves into the following steps is considerably reduced, requiring less computational power but also reducing the density of the final 3D reconstruction [21]. Indirect methods typically perform their optimization steps by minimizing the reprojection error due to the feature type of information that the preprocessing step outputs [41]. On the other hand, direct methods work directly on the pixel intensity information without requiring preprocessing steps, implying that the algorithm has more information for estimation tasks allowing one to obtain denser reconstructions of the scene, requiring more computational power [41]. In addition, direct methods typically perform their optimization steps based on the photometric error due to the direct pixel availability information.

- **Dense vs. sparse**. Dense vs. sparse classification refers to the amount of information recovered in the final map as a 3D reconstruction [21]. Denser reconstructions have more definition and continuity in the reconstructed objects and surfaces. In contrast, sparser reconstructions are typically represented as largely separated point clouds, where the edges and corners are commonly the only objects that can be recognized clearly [7].

- **Classic vs. machine learning.** Classic methods have been proposed in the last three decades, typically basing their formulation on geometric, optimization, and probabilistic techniques without machine learning. However, in recent years, due to the impressive advances in artificial intelligence, especially in Convolutional Neural Networks (CNN), many techniques have been applied to improve the SLAM or VO estimation tasks [11,32,36,42]. The methods based on classic formulations enhanced with machine learning are called Machine-learning-based approaches (ML).

Combining these three classifications in all their possible configurations [9] establishes the taxonomy: Classic + Dense + Direct, Classic + Sparse + Direct, Classic + Dense + Indirect, Classic + Sparse + Indirect, Classic + Hybrid, ML + Dense + Direct, ML + Sparse + Direct, ML + Dense + Indirect, ML + Classic + Sparse + Indirect, and ML + Hybrid. It must be mentioned that the hybrid category was added for those methods that efficiently combine the direct and indirect principles to estimate ego-motion and scene geometry, like SVO [13] and CNN-SVO [11]. Figure 1 depicts the monocular RGB taxonomy for the SLAM, SFM, and VO algorithms.

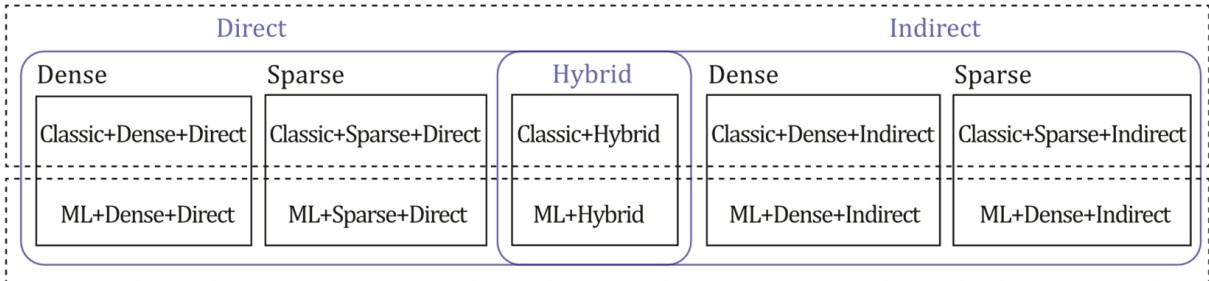

**Figure 1.** Diagram of the SLAM, SFM, and VO taxonomy. Inspired by [9].

### 2.2. Selected Algorithms

In this comparative analysis, we aimed to determine the taxonomy classifications, limitations, and advantages by exploring as many taxonomy categories as possible. For this purpose, we selected and implemented five methods of geometric-based classification. Furthermore, we included three machine-learning versions of the selected classic approaches to test the hypothesis of whether or not the addition of a CNN to classic approaches significantly improved the geometric-based methods' performance. In a previous work [9], many machine learning approaches were listed available as open-source code [17,19,42–53]. However, during their implementation, we found that many implementations were avail-

able for multiple input modalities like RBD-D or INS. However, the provided code was not available for monocular RGB as the unique input source of information, or they required external and not included modules for their implementation, e.g., [42,43,50,53]; thus, we could not include those methods for this comparison. Finally, we added two additional sparse-direct implementations built over the DSO [21] system, given the impressive 3D reconstruction results that this method demonstrated during evaluations.

In this way, the algorithms selected to perform this comparative study were:

1.  **ORB-SLAM2.** As a sparse-indirect representative, we selected ORB-SLAM2 [54], widely known as the gold standard of this category, as most of the currently available sparse-indirect methods are proposals inspired by this algorithm. The original ORB-SLAM [24] extracts ORB features as preprocessing of multiscale FAST corners with a 256-bit descriptor giving that algorithm information to perform a Bundle Adjustment for optimization and work in three threads for tracking, local mapping, and loop closure. In addition, the ORB-SLAM2 incorporates a fourth thread to perform full Bundle Adjustment after loop closure extending the original method and obtaining the scene optimal geometric representation. The ORB-SLAM2 is publicly available as an open source code in [55]; it may be implemented in its C++ version or ROS version, with minimum additional requirements, Pangolin, OpenCV (tested for 2.4.3 version), Eigen 3 (tested for 3.1.0 version), DBoW2, and g2o, which are included in the repository.

2.  **DF-ORB-SLAM**. Classic dense-indirect methods available in the literature, like [25,26], are not available as open-source code for implementation; so, they could not be considered for this evaluation. Instead, a well-known classic dense-direct version of ORB-SLAM2 exists, called DF-ORB-SLAM [16], with its code publicly available on GitHub. The DF-ORB-SLAM algorithm was built based on the ORB-SLAM2 algorithm, allowing the addition of depth map retrieval capabilities and incorporating optical flow to track the detected points; thus, this algorithm uses a large amount of information obtained through the input using most of the pixel values for optical flow estimation. Once the optical flow is estimated, the ORB-SLAM2 performs feature extraction on the optical flow domain executing the rest of its pipeline. The DF-ORB-SLAM is publicly available in [16], implemented in Ubuntu 18.04 in its ROS version using its official *build_ros.sh* script.

3.  **LSD-SLAM.** The LSD-SLAM [29] is one of the most popular methods of the dense-direct category, since it has been the basis and inspiration for a lot of the methods currently available [10,21,56]. The LSD-SLAM not only locally tracks the camera's movement but also allows the construction of dense maps through a semi-dense geometric representation tracking the depth values only in high-gradient areas. The method has direct image alignment mechanisms and estimation based on the semi-dense depth map filtering technique [57]. The global depth map is rendered as a pose graph comprising keyframes represented as vertices that present feature 3D similarity transformations as edges, adding environment scaling ability and allowing the accumulated drift to be detected and corrected. Furthermore, the LSD-SLAM uses an appearance-based loop detection algorithm called FAB-MAP [58], introducing prominent loop closure candidates that extract their features without reusing any additional visual odometry information. The LSD-SLAM is publicly available in [59] and was implemented in Ubuntu 18.04 in its ROS version.

4.  **DSO**. The DSO [21] is widely known as the direct methods' gold standard due to the impressive reconstruction and odometry results that it has achieved, inspiring other implementations and new proposals. The DSO works directly on the pixel intensity information but applies a point selection strategy to reduce the amount of information to be processed efficiently, continuously optimizing the photometric error applied to the last N-frames while optimizing the complete likelihood for the parameters involved in the model, including poses, intrinsics, extrinsics, and inverse depths, executing a windowed sparse bundle adjustment. The DSO is publicly available for

implementation in [60]; its code runs entirely in C++, using minor requirements like Suitesparse, Eigen3, and Pangolin.

5.  **SVO.** We selected the most commonly known method, SVO [12], for the hybrid classification. The SVO efficiently combines the advantages of the direct and indirect approaches by using the feature correspondences obtained on the direct motion estimation for tracking and mapping. This procedure considerably reduces the number of required features and is only executed when a new keyframe is selected to insert new points in the map. First, camera motion is estimated by a sparse model-based image alignment algorithm, where sparse point features are used to estimate the feature correspondences. Next, this information is used to minimize the photometric error. Then the reprojected points, pose, and structure are refined by minimizing the reprojection error. The SVO is publicly available in [61] for testing and implementation running on C++ or ROS. Modern operating systems might find issues during implementation; so, Ubuntu 16.04 and ROS kinetic were used.

6.  **LDSO.** As an additional sparse direct system, the LDSO [30] was selected as an update of the DSO algorithm that includes loop-closure capabilities. The LDSO enables the DSO framework to detect the loop closure by ensuring point repeatability using corner features to detect loop candidates. For this purpose, the depth estimates for point features allow the algorithm to compute the $Sim(3)$ constraints, to be combined with the pose-only bundle adjustment and point cloud alignment and fused with the relative pose DSO covisibility graph, sliding the window optimization stage. This way, the LDSO adds the loop closure to the DSO system, including a loop closure module based on a global pose graph optimization working over the last five to seven keyframes' sliding window. The LDSO was made publicly available in [62], and for this comparison, it was implemented in Ubuntu 18.04 along with OpenCV 2.4.3, Sophus, DBoW3, and g2o.

7.  **DSM**. Another sparse-direct method we were interested in testing was the DSM [31], another update made to the DSO to create a complete SLAM system. The DSM aimed to include scene reobservation information to enhance the precision and reduce the drift and inconsistencies. In contrast to the LDSO, which considers a sparse set of reobservations, the DSM builds a persistent map allowing the algorithm to reuse existing information by a photometric formulation. The DSM uses local map covisibility window criteria to detect the active keyframes reobserving the same region, a coarse-to-fine strategy to process that point reobservation information and a robust nonlinear photometric bundle adjustment technique based on the photometric error for outlier detection. The DSM open-source code is publicly available in [63], which was implemented for comparisons on Ubuntu 18.04 with Eigen (v3.1.0), OpenCV (v2.4.3), and Ceres solver, which were provided in the official repository.

8.  **DynaSLAM**. The Dyna-SLAM algorithm [32] is a lighter version of ORB-SLAM2 exceeded by adding the detection, segmentation, and inpainting of dynamic information on scenes' machine learning capabilities. In addition, the Mask R-CNN of [64] was integrated with the classic sparse-indirect method to detect and segment regions of each image that potentially belonged to movable objects. The authors also incorporated a multiview geometry approach calculating backprojections to define the key point parallax angles to detect additional information the CNN cannot recognize. The authors reported that this combination contributed to overcoming the ORB-SLAM2 initialization issues; so, it works in dynamic environments. The DynaSLAM is publicly available in [17], and it was implemented in Ubuntu 16.04 with ROS Kinetic, Cuda 9, Tensorflow 1.4.0, and Keras 2.0.8.

9.  **CNN-DSO**. In the literature, DSO neuronal methods like D3VO [65], MonoRec [14], and DDSO [66] can be found. Nevertheless, they are not publicly available, or in the case of MonoRec, its monocular VO pipeline is not available for testing; so, the CNN-DSO was selected for this comparison, which is publicly available in [15]. This method includes a CNN depth prediction module enabling the DSO system to execute

its estimation modules using additional depth prior information obtained by the network. The CNN used for this study was the MonoDepth system of [67], a single image depth estimation network that outputs a depth value for each pixel position by chains of feature maps processing. The network was built over the ResNet backbone using a variant of its encoder–decoder architecture. The CNN-DSO requires building TensorFlow (v1.6.0) from source and MonoDepth from its official repository [68], and it was implemented in Ubuntu 18.04, with Eigen (v3.1.0) and OpenCV (v2.4.3).

10. **CNN-SVO**. In the study of [11], an extension of the hybrid method SVO was proposed by fusing the same Single Image Depth Estimation (SIDE) CNN MonoDepth module used in the CNN-DSO with the original geometric-based hybrid method. In this case, MonoDepth was included to add preliminary depth information to the SVO pipeline, minimizing the uncertainty in the feature correspondence identification steps; then, the system is initialized, obtaining high uncertainty maps. Then, the SIDE CNN creates filters to approximate the current values' mean and variance, considerably reducing the amount of information separating inliers/outliers in the depth map. The CNN-SVO is publicly available in [46] and was implemented in Ubuntu 16.04 to allow the SVO modules to work with ROS Kinetic.

For more information on the taxonomy, definitions, SLAM, VO, and SFM basics, and further details of the methods described in this review, we encourage the reader to see the prior works listed in [9].

### 2.3. Benchmarks

Today, the scientific community has considerably promoted the development of datasets, including existing open-source datasets even for evaluating complex hardware setups like visual–inertial systems (i.e., YTU [69], WHUVID [70], and VOID [71]). In this way, there are several datasets and benchmarks available in the literature for evaluating RGB SLAM, SFM, and VO systems, like [6,8,72–80]. Nevertheless, only a few are suitable for pure monocular RGB systems due to the nature of image acquisition, the type of camera calibration or camera models used, and the format of the provided ground truth. Similarly, it is safe to say that among the reviewed available datasets, the following can be applied for monocular algorithms comparison:

- The **KITTI** dataset in [74] contains 21 video sequences of a driving car, where the movement parameters are limited to forward driving. The available images have pre-rectification treatments, and the dataset provides a ground truth obtained through an assembly of GPS and INS.

- The **EUROC-MAV** dataset in [75] contains 11 inertial stereoscopic sequences of a quadcopter flying in different indoor environments providing groundtruth values of all frames and calibration parameters.

- The **TUM-Mono** dataset in [6] presents 50 sequences of indoor/outdoor environments obtained using monocular RBG cameras on monochrome uEye UI-3241LE-M-GL cameras equipped with Lensagon BM2420 (with $148° \times 122°$ field of view) and Lensagon BM4018S118 (with $98° \times 79°$ field of view) sensors. This benchmark includes the photometric calibration parameters, the ground truth, the timestamps for the execution of each image sequence, and the calibration file for the vignetting effect in each sequence, comprising more than 190,000 frames and more than 100 min of video.

- The **ICL-NUIM** benchmark in [8] has eight sequences in conjunction with its ray-tracing of two environments, providing the groundtruth values of each sequence and camera intrinsics; so, no photometric calibration is required. This dataset presents degenerative and purely rotational motion sequences, which are considered demanding for monocular algorithms.

As can be noticed, the most complete and largest dataset of the above is the TUM-Mono, which is why this dataset was applied in this comparison study. It also has the advantage of being the only dataset that was obtained purely depending on a monocular

RGB setup, without depending on any additional sensor or source of information as mentioned in [6,39,40], making it ideal for comparing visual-only SLAM and VO systems. In addition, this benchmark provides the most complete set of metrics that can be explored to efficiently compare the selected algorithms in multiple dimensions—discussed in the following section.

*2.4. Metrics*

As SLAM, SFM, and VO are ill-posed problems that can be addressed from multiple perspectives and a wide variety of techniques, comparing the final obtained 3D reconstruction is not the best alternative for monocular RGB methods because of the different sparsity, scale, and type of output that each method brings, due to the difficulty of accruing accurate groundtruth maps [81]. At the same time, trajectories can be acquired using INS, GPS, LASER, RADAR, LIDAR, and Kinect systems, among others, with acceptable accuracy. In this way, as discussed in [6], the best way of comparing SLAM and VO algorithms of diverse nature (see Figures 1 and 2) is by comparing the output trajectory in each algorithm, because even if the method is focused on reconstruction only, it has been demonstrated that solving both problems of landing and mapping simultaneously brings the best reconstruction results [3], as the quality of the final reconstruction tightly depends in the quality of the ego-motion estimation. Hence, the metrics we used for this comparison are entirely based on ego-motion estimation, which can be effectively compared for all SLAM and VO algorithms. Among the different metrics for ego-motion available in the literature, we found that the metrics present in most of the methods listed in [9] were: the absolute trajectory RMSE (ATE), the relative pose RMSE (RPE), the cumulated trajectory, rotation, and scale errors, the alignment error, and the alignment RMSE.

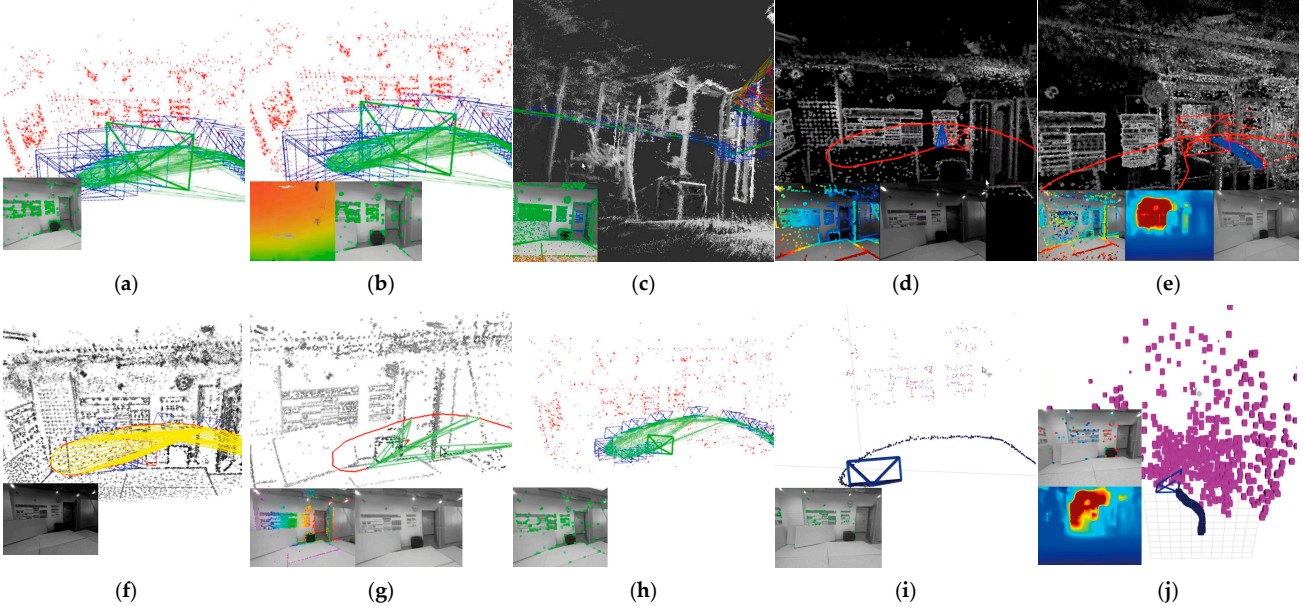

**Figure 2.** Examples of each indoor algorithm sequence execution *seq-01*—TUM-Mono dataset. The implemented methods were: (**a**) ORB-SLAM2, (**b**) DF-ORB-SLAM, (**c**) LSD-SLAM, (**d**) DSO, (**e**) CNN-DSO, (**f**) LDSO, (**g**) DSM, (**h**) DynaSLAM, (**i**) SVO, and (**j**) CNN-SVO.

The ATE and RPE are local pose accuracy metrics proposed by [81], commonly applied along with the EUROC dataset. The relative pose error is a metric for the accuracy of an estimated trajectory over a defined time interval Δ. In this way, this metric corresponds to the drift of the estimated trajectory. For a sequence of estimated poses $P_1, \dots, P_n \in SE(3)$

and a ground truth trajectory $Q_1, \ldots, Q_n \in SE(3)$, the relative pose error for each timestamp $i$ is:

$$E_i := \left(Q_i^{-1} Q_{i+\Delta}\right)^{-1} \left(P_i^{-1} P_{i+\Delta}\right) \tag{1}$$

So, for a sequence of n poses, an $m = n - \Delta$ number of relative poses is obtained. Then the root mean square error (*RMSE*) for such errors over all the timestamps of the sequence can be calculated as:

$$RMSE(E_{i:n}, \Delta) := \sqrt{\frac{1}{m} \sum_{i=1}^{m} \|trans(E_i)\|^2} \tag{2}$$

where $trans(E_i)$ corresponds to the translational component of the relative pose error $E_i$. Many VO or SLAM systems can be evaluated for a timestep interval $\Delta = 1$, but some methods work on frames or keyframes windows (like [21,30,63]); thus, different $\Delta$ values might be appropriate for testing. So, for SLAM systems' evaluation, it can also be useful to obtain the *RMSE* for all the possible time intervals:

$$RMSE(E_{i:n}) = \frac{1}{n} \sum_{\Delta=1}^{n} RMSE(E_{i:n}, \Delta) \tag{3}$$

The ATE was proposed to evaluate the estimated trajectory's global consistency. The ATE estimation was achieved by comparing the absolute distances between the estimated trajectory and the ground truth directly. So, the trajectories are first aligned using Horn's method [82] to find the rigid body transformation $S$ to map the estimated trajectory $P_{1:n}$ into the ground truth trajectory $Q_{1:n}$; hence, the absolute trajectory error for each timestamp $i$ can be calculated as:

$$F_i := Q_i^{-1} S P_i \tag{4}$$

In the same way as RPE, the *RMSE* for all the timestamps of the translational components is calculated as follows:

$$RMSE(F_{i:n}) := \sqrt{\frac{1}{n} \sum_{i=1}^{n} \|trans(F_i)\|^2} \tag{5}$$

Thus, for the [81] benchmark, the RPE combines the translational and rotational errors elegantly, while the ATE considers only the translational error component. In contrast, for the TUM-Mono benchmark [6], the authors proposed to benefit from large loop sequences. This way, instead of using the complete exploring motion pose information, the TUM-Mono benchmark was built to register the ground truth of each sequence's first and last 10–20 s, using the LSD-SLAM [29] method to track only those segments. In this way, the authors used the accumulated drift for all their metrics, and they demonstrated that the error registered by each evaluated run was not originated in the SLAM method drift used to register the ground truth; instead, it came from the accumulated drift by the algorithm through the entire trajectory. So, any SLAM or VO system can be used to register the start and end segments' ground truth. Consequently, the TUM-Mono benchmark aligns the estimated trajectory with the start and end ground truth segments and measures their differences. Let $p_1, \ldots, p_n \in \mathbb{R}^3$ be the estimated tracked positions for the 1 to $n$ frames and $S \subset [1; n]$ and $E \subset [1; n]$ be the frame indices corresponding to the start and end segments for the groundtruth positions $\hat{p} \in \mathbb{R}^3$. Then, by aligning the estimated trajectory with the groundtruth start and end segments, the two relative transformations can be calculated as:

$$T_s^{gt} := \underset{T \in Sim(3)}{\arg\min} \sum_{i \in S} (T p_i - \hat{p}_i)^2 \tag{6}$$

$$T_e^{gt} := \underset{T \in Sim(3)}{\arg\min} \sum_{i \in E} (T p_i - \hat{p}_i)^2 \tag{7}$$

By these transformations, the accumulated drift can be calculated as:

$$T_{drift} := T_e^{gt} \left( T_e^{gt} \right)^{-1} \tag{8}$$

The translation, rotation, and scale error components can be extracted as, respectively:

$$e_t := \left\| translation \left( T_{drift} \right) \right\|$$

$$e_r := rotation \left( T_{drift} \right)$$

$$e_s := scale \left( T_{drift} \right)$$

As a result, the authors established the alignment error, which is a metric that equally takes into account the errors produced by the translational, rotational, and scale effects:

$$e_{align} := \sqrt{\frac{1}{n} \sum_{i=1}^{n} \left\| T_s^{gt} p_i - T_e^{gt} p_i \right\|_2^2} \tag{9}$$

This metric can be computed individually for the start and end segment, but when it is estimated by combining both intervals, it is equivalent to the translational RMSE when aligned to the ground truth. Thus, it can also be formulated as follows:

$$e_{rmse} := \sqrt{\min_{T \in Sim(3)} \frac{1}{|S \cup E|} \sum_{i \in S \cup E} (T p_i - \hat{p}_i)^2} \tag{10}$$

As can be observed, in contrast to the APE and ATE, which only include two metrics explaining the rotation and translation effects, the TUM-Mono benchmark analyzes the SLAM or VO performance method in a more detailed way, providing six metrics explaining the amount of the accumulated translation, rotation, and scale errors, as well as determining the performance of the algorithm in the start and end segment to better identify the initialization and accumulated drift errors, finally calculating the translational RMSE to visualize the global effects of the combined metrics on the whole evaluated sequence. For these reasons, we selected the TUM-Mono and its official metrics to execute a complete comparison.

## 3. Results

As mentioned above, the algorithms were selected based on their open-source availability and independence from any additional input information source other than a monocular RGB frame sequence. Following the primary taxonomy described in [9,21], we selected the classic sparse-indirect system ORB-SLAM2 [54], the classic dense-indirect system DF-ORB-SLAM [16], the classic dense-direct system LDS-SLAM [29], and the classic sparse-direct method DSO [21]. Then, we added to this study the currently available ML implementations of the ORB-SLAM2 [54], DSO [21], and SVO [13], which are the DynaSLAM [32], CNN-DSO [15], and CNN-SVO [11]. Additionally, the direct proposals derived from the DSO system were included due to the impressive reconstruction results that this method showed during experimental evaluation; thus, the LDSO [30] and DSM [31] systems representing the addition of loop closure and SLAM capabilities for the DSO system were selected. Figures 2 and 3 present some examples of the evaluated algorithms' execution on the outdoor and indoor sequences of the TUM-Mono dataset.

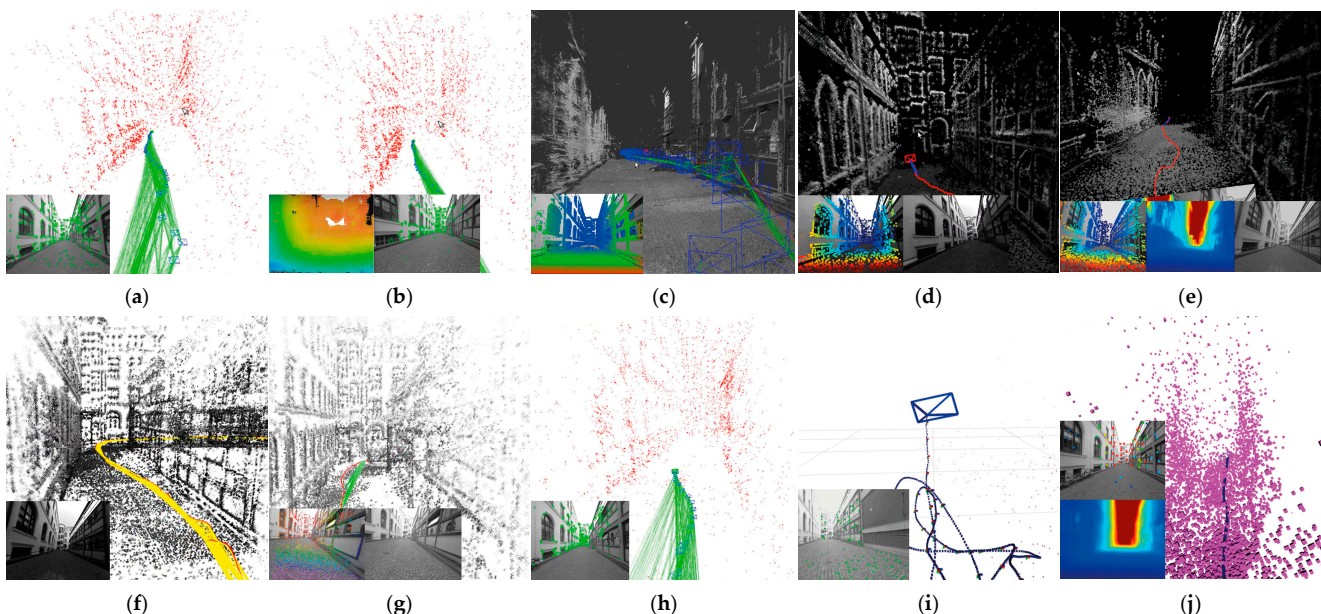

**Figure 3.** Algorithm executions for the outdoor sequence *seq-02* of the TUM-Mono dataset. The implemented methods were: (**a**) ORB-SLAM2, (**b**) DF-ORB-SLAM, (**c**) LSD-SLAM, (**d**) DSO, (**e**) CNN-DSO, (**f**) LDSO, (**g**) DSM, (**h**) DynaSLAM, (**i**) SVO and (**j**) CNN-SVO.

### 3.1. Hardware Setup

All the methods were evaluated on the same hardware platform with the same available computational and power resources using Ubuntu 18.04 and 16.04 operating systems, depending on each algorithm's software requirements. For this evaluation, we selected readily available and cheap hardware components to assemble a desktop based on the AMD Ryzen™ 7 3800X processor and the GPU NVIDIA GEFORCE GTX 1080 Ti. The technical specifications are summarized in Table 1.

**Table 1.** Specifications of the hardware used during experimentation.

| Component | Specifications |
| --- | --- |
| CPU | AMD Ryzen™ 7 3800X, eight cores, 16 threads, 3.9–4.5 GHz. |
| GPU | NVIDIA GEFORCE GTX 1080 Ti. Pascal architecture, 1582 MHz, 3584 CUDA cores, 11 GB GDDR5X. |
| RAM | 16 GB, DDR 4, 3200 MHz |
| ROM | SSD NVME M.2 Western Digital 7300 MB/s |
| Power consumption | 750 W [1] |

[1] The hardware did not reach the max power consumption. The avg. load was close to 600 W during the experiments.

Additionally, we provide some CPU performance metrics obtained by averaging the metrics of ten executions of each algorithm over the *sequence*_01 of the TUM-Mono dataset. All these CPU usage metrics were obtained using the hardware detailed in Table 1, using the official codes provided in their repositories and the same dependencies' versions listed by each author. The metrics selected to provide an idea of the computational expenses required by each algorithm were: the overall CPU usage (multicore), the current CPU usage, the amount of GPU memory required by the algorithm, the amount of RAM used while running the algorithm, the time that the algorithm required to process the *sequence*_01 of the TUM-Mono dataset, and the number of frames per second that the algorithm processed. The computational expenses associated with each approach to reconstruct the *sequence*_01 scene are presented in Table 2.

**Table 2.** The average CPU usage performance metrics for the ten executions of each algorithm on *sequence_*01 of the TUM-Mono dataset.

| Method | Overall CPU Usage, Multicore | Current CPU Usage | GPU Usage | Memory Usage | Time (s) | FPS |
|---|---|---|---|---|---|---|
| ORB-SLAM2 | 1.8472% | 16.2374% | 1.2376% | 9.2147% | 128.4571 | 37 |
| DF-ORB-SLAM | 1.9254% | 17.4235% | 1.7861% | 12.4572% | 133.1217 | 36 |
| LSD-SLAM | 2.4578% | 18.4521% | 1.6423% | 10.3457% | 138.4172 | 34 |
| DSO | 1.2604% | 14.6818% | 1.8971% | 9.3892% | 91.2564 | 52 |
| SVO | **1.1286%** | **10.4589%** | **1.7852%** | **8.5316%** | **87.5241** | **55** |
| LDSO | 1.6909% | 15.4717% | 3.1588% | 14.2962% | 99.4758 | 48 |
| DSM | 1.8346% | 31.9216% | 2.7203% | 24.1591% | 315.4982 | 15 |
| DynaSLAM | 1.9247% | 21.4576% | 15.3467% | 20.3879% | 118.3245 | 40 |
| CNN-DSO | 4.0647% | 30.9091% | 27.5346% | 24.6742% | 161.2389 | 30 |
| CNN-SVO | 3.2579% | 27.8461% | 24.4732% | 23.5476% | 134.7583 | 35 |

As can be noticed in Table 1, the SVO was the fastest algorithm that required fewer computational resources to be implemented, closely followed by the DSO and ORB-SLAM2. However, as described in the following sections, the SVO presented strong trajectory loss issues and poor 3D reconstruction quality; so, it might not be considered the best alternative. In addition, it can be noticed that adding ML techniques to geometric-based approaches implied a considerable increase in the CPU, GPU, and memory use.

### 3.2. Comparative Analysis

As mentioned in Section 2.3, we used the TUM-Mono dataset and benchmark because it has a complete set of metrics; all its sequences were gathered using only monocular cameras, and it presents the largest collection of 50 sequences and scenarios that comprise multiple outdoor and indoor examples. In addition, it must be considered that these sequences were gathered using a pure monocular handheld camera and were recorded by a walking person; so, the results presented in this section might not be generalized to considerably different applications like autonomous driving, flying drones, and medical exploration applications, among others.

As addressed in the related works [6,54], we followed the authors' suggestions of running each sequence of a benchmark five times to create cumulative-error plots and account for the nondeterministic nature of each system [40]. Nevertheless, authors like [30,32] performed their experimental comparisons by running each sequence ten times in forward and backward reproduction directions to better capture the probabilistic behavior of the algorithms against multiple variations like illumination and dynamic objects. In this way, we applied this extended approach, given the large variety of algorithms we tested. In total, we performed ten runs of each of the 50 sequences in forward and backward modalities, gathering a total of 1000 runs for each method; so, for the ten evaluated algorithms, we created a database of 10,000 trajectory files saved in .txt format that were processed using the MATLAB scripts provided in the official repository of the TUM-Mono benchmark [6].

The TUM-Mono benchmark scripts require a specific structure, where each algorithm must output a .txt file containing all the camera poses registered during each sequence algorithm execution, where each $P_i$ pose must be in the quaternion format represented in Equation (1). However, the SVO, CNN-SVO, and DSM algorithms output rotation and translation matrixes instead of quaternions, which are incompatible with the TUM-Mono format. In this way, we had to modify the codes of these methods to introduce the correct output format, following the proposal of Sarabandi and Thomas [83], by applying Equations (11)–(15) to convert the translation and rotation outputs of the SVO, CNN-SVO, and DSM to quaternions.

$$P_i = \left( t_i \; x_i \; y_i \; z_i \; q_{x_i} \; q_{y_i} \; q_{z_i} \; q_{w_i} \right) \tag{11}$$

Given the rotation matrix:

$$\boldsymbol{R} = \begin{pmatrix} r_{11} & r_{12} & r_{13} \\ r_{21} & r_{22} & r_{23} \\ r_{31} & r_{32} & r_{33} \end{pmatrix}$$

$$q_x = \begin{cases} \dfrac{1}{2}\sqrt{1 + r_{11} + r_{22} + r_{33}}, & if\ r_{11} + r_{22} + r_{33} > \eta \\ \dfrac{1}{2}\sqrt{\dfrac{(r_{32} - r_{23})^2 + (r_{13} - r_{31})^2 + (r_{21} - r_{12})^2}{3 - r_{11} - r_{22} - r_{33}}}, & otherwise \end{cases} \tag{12}$$

$$q_y = \begin{cases} \dfrac{1}{2}\sqrt{1 + r_{11} - r_{22} - r_{33}}, & if\ r_{11} - r_{22} - r_{33} > \eta \\ \dfrac{1}{2}\sqrt{\dfrac{(r_{32} - r_{23})^2 + (r_{12} + r_{21})^2 + (r_{31} + r_{13})^2}{3 - r_{11} + r_{22} + r_{33}}}, & otherwise \end{cases} \tag{13}$$

$$q_z = \begin{cases} \dfrac{1}{2}\sqrt{1 - r_{11} + r_{22} - r_{33}}, & if\ -r_{11} + r_{22} - r_{33} > \eta \\ \dfrac{1}{2}\sqrt{\dfrac{(r_{13} - r_{31})^2 + (r_{12} + r_{21})^2 + (r_{23} + r_{32})^2}{3 + r_{11} - r_{22} + r_{33}}}, & otherwise \end{cases} \tag{14}$$

$$q_z = \begin{cases} \dfrac{1}{2}\sqrt{1 - r_{11} - r_{22} + r_{33}}, & if\ -r_{11} - r_{22} + r_{33} > \eta \\ \dfrac{1}{2}\sqrt{\dfrac{(r_{21} - r_{12})^2 + (r_{31} + r_{13})^2 + (r_{32} + r_{23})^2}{3 + r_{11} + r_{22} - r_{33}}}, & otherwise \end{cases} \tag{15}$$

As reported in [83], the best results that outperformed Shepperd's rotation to the quaternion method were achieved for $\eta = 0$; so, we set this value to build the trajectory files for those methods that did not match the evaluation format. In addition, the ORB-SLAM2, DF-ORB-SLAM, DynaSLAM, SVO, CNN-SVO, and DSM required a different calibration camera model than that the provided by the benchmark that includes full photometric data considering the geometric intrinsic calibration, photometric calibration, and the nonparametric vignette calibration, while the rest of the methods used an ATAN camera model based on the FOV distortion model of [84] provided in the official PTAM repository [85]. We used the ROS calibration package [86] to estimate three radial and two tangential distortion coefficients $\boldsymbol{d}_{coeff} = (k_1\ k_2\ p_1\ p_2\ k_3)$, following the formulation of Equations (6) and (7). The results were also tested and compared with the OpenCV Camera calibration package [87].

For each undistorted pixel at $(x_u, y_u)$ coordinates, its position in the distorted image is $(x_d, y_d)$:

$$\begin{aligned} x_u &= x_d\left(1 + k_1 r^2 + k_2 r^4 + k_3 r^6\right) \\ y_u &= y_d\left(1 + k_1 r^2 + k_2 r^4 + k_3 r^6\right) \end{aligned} \tag{16}$$

$$\begin{aligned} x_u &= x_d + \left[2p_1 x_d y_d + p_2\left(r^2 + 2x_d^2\right)\right] \\ y_u &= y_d + \left[p_1\left(r^2 + 2y_d^2\right) + 2p_2 x_d y_d\right], \end{aligned} \tag{17}$$

where $r$ is the distorted radius $r_d = \sqrt{x_d^2 + y_d^2}$. As suggested in [6], to make a fair comparison based on the accumulated drift over the aligned start and end sequences, we disabled the loop closure for the SLAM methods ORB-SLAM2, DF-ORB-SLAM, DynaSLAM, LDSO, and DSM. Figure 4 presents each algorithm's cumulative error plots for the translational, rotational, and scale errors. These graphs depict the number of runs for each error type

below a certain x-value. Hence, the methods close to the top left corner were better because they reached a determined error value after more executions.

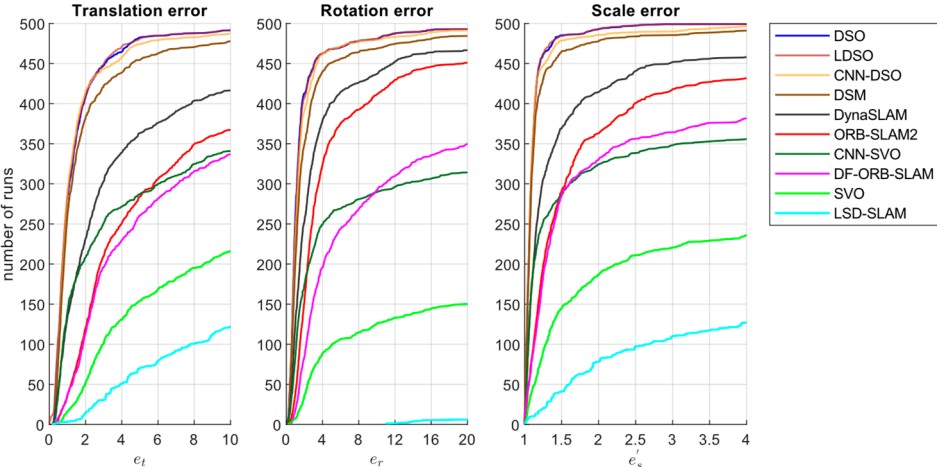

**Figure 4.** Translation $e_t$, rotation $e_r$, and scale $e'_s$ accumulated errors for the ten evaluated algorithms.

As can be seen in Figure 4, the sparse-direct methods (DSO, LDSO, and DSM) achieved the best overall performance, followed by the sparse-indirect method (ORB-SLAM2), the dense-indirect method (DF-ORB-SLAM), and the hybrid method (SVO); the dense-direct method (LDS-SLAM) showed the worst performance. The ORB-SLAM2 and SVO CNN versions showed an important improvement over their classic versions. At the same time, the CNN-DSO did not outperform the DSO in accumulated translation, rotation, and scale metrics but remained close to the performance of the DSO. Finally, it must be mentioned that the large error observed in the LSD-SLAM, SVO, and CNN-SVO methods can be attributed to the severe initialization and relocalization problems that the algorithms presented during the evaluations in the TUM-Mono dataset.

As mentioned, the alignment error considers the translation, rotation, and scale errors equally. Therefore, it is equivalent to the translational RMSE when aligned to the start and end segments (the first and last 10–20 s of each sequence), for which the ground truth is available. The cumulated alignment error for each algorithm is presented in Figure 5.

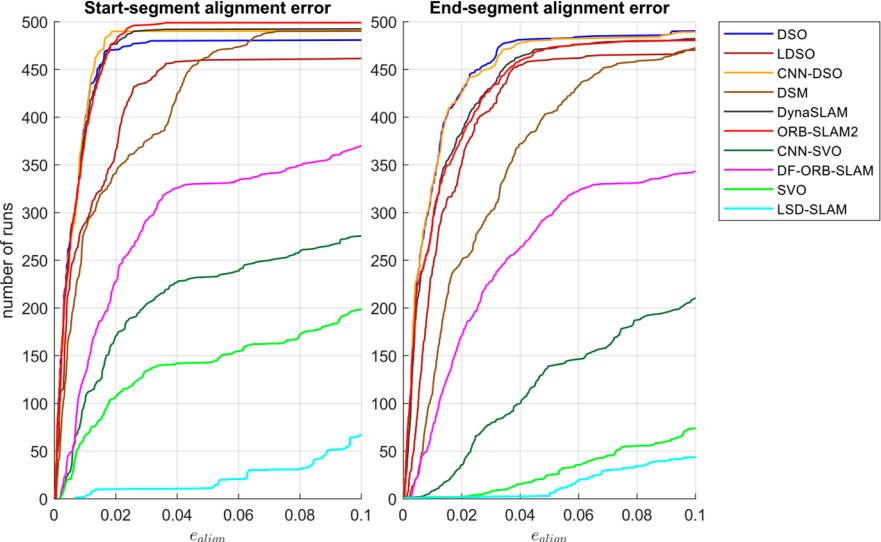

**Figure 5.** Start and end segment alignment error, corresponding to the RMSE of the alignment error when compared with the start and end segment's ground truth.

Figure 5 shows that the ORB-SLAM and DynaSLAM performed slightly better than the sparse direct methods for start-segment alignment errors. However, on the end segment, the cumulative drift effect was lower on the sparse-direct methods ratifying the results observed in Figure 4. In addition, it can be noticed that the CNN-DSO performed better than the DSO, suggesting that integrating the Single Image Depth Estimation (SIDE) CNN improved the DSO bootstrapping by adding the prior depth information, whereas the end-segment performance of both algorithms was similar. Moreover, the addition of the Mask R-CNN in DynaSLAM was used to remove scenes' moving objects information and did not represent a clear improvement in algorithm performance in the start segment, but, as shown in Figure 5, the benefits of adding the CNN can be observed over the end segment by the reduction in the accumulated drift. Additionally, for the hybrid approaches, the addition of the MonoDepth CNN made a paramount contribution in helping to overcome SVO loss of trajectory issues. Similar to the results in Figure 1, the overall alignment error results suggest that the sparse-direct methods performed better, followed by the sparse-indirect, dense-indirect, hybrid, and finally the dense-direct, reaching a threshold error in around 50 runs.

As suggested by [6], we examined the dataset motion bias for each algorithm by running each method ten times forwards and ten times backward in such modalities and combining both to visualize how much each algorithm is affected. This situation allowed us to consider the importance of evaluating the SLAM and VO methods in large datasets, covering as many environments and motion patterns as possible. The dataset motion bias for each method is presented in Figure 6.

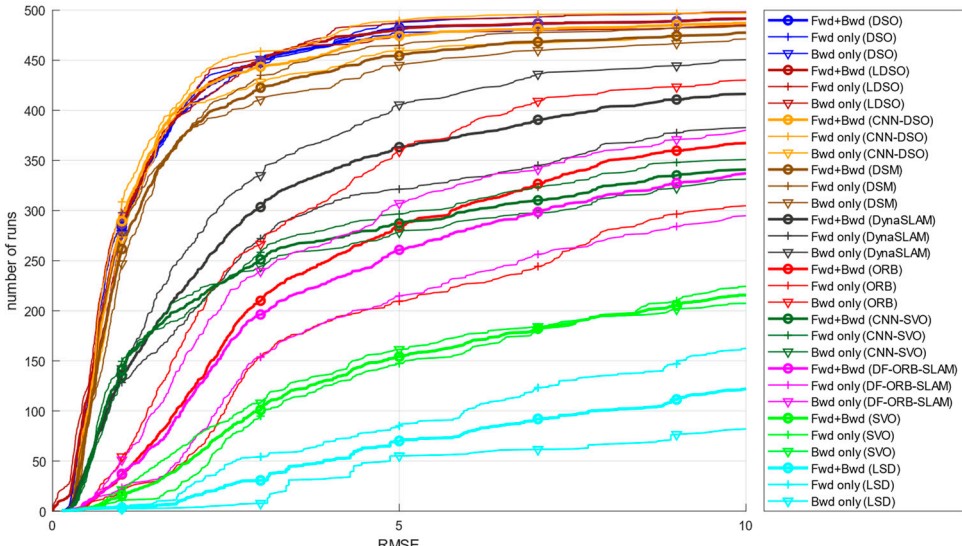

**Figure 6.** The dataset motion bias for each method was evaluated by running all sequences forwards and backward, as well as their combination (bold).

In Figure 6, it can be noticed that the DSO, LDSO, and SVO were not seriously affected by motion bias. In contrast, different motion patterns considerably affected the ORB-SLAM2, DynaSLAM, and DF-ORB-SLAM. This can be observed in the performance differences when running forwards versus backward. This behavior provides a reference for the consistency and robustness of each algorithm for using them in different environments or applications. It can be observed that the CNN-DSO on forward-only modality outperformed its classic version, but it suffered from a larger motion bias effect affecting its overall performance; while DynaSLAM and CNN-SVO outperformed their classic versions and presented less motion bias effect, representing an additional robustness improvement over their classic versions.

Figure 7 shows the color-coded alignment error for each of the 50 TUM-Mono sequences for each run forward and backward to observe which specific sequences were challenging for each algorithm.

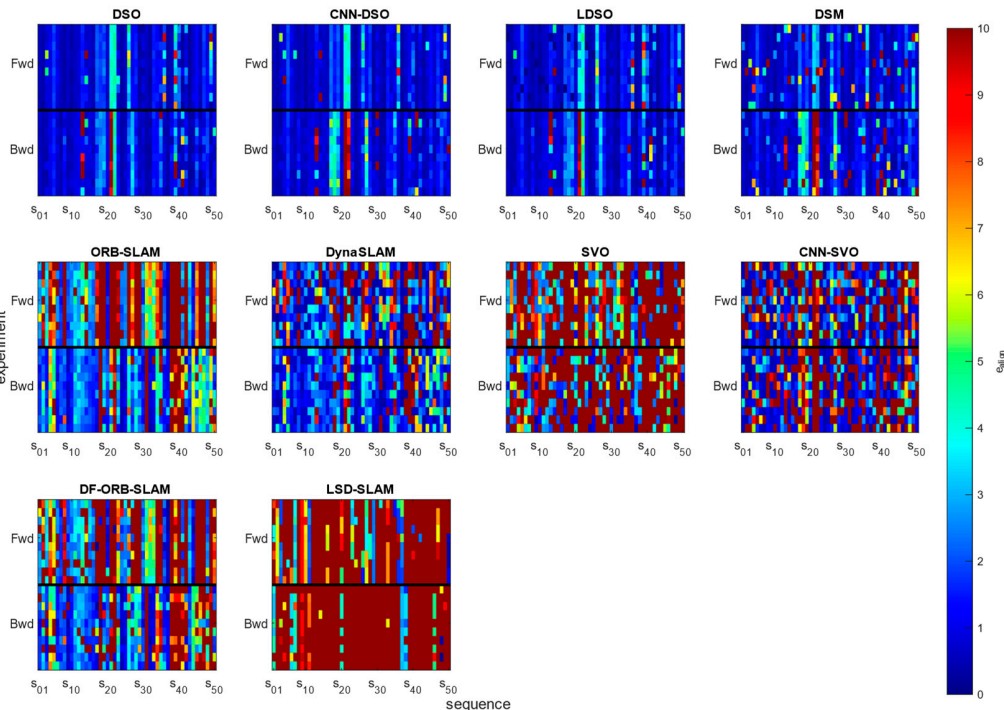

**Figure 7.** The color-coded alignment error $e_{align}$ for each algorithm in the TUM-Mono dataset.

The first row of Figure 7 presents the sparse-direct methods, DSO, CNN-DSO, LDSO, and DSM, demonstrating an outstanding performance compared to the rest of the evaluated methods belonging to different taxonomy classifications placing the sparse-direct methods as the best alternative for the Visual Odometry, SLAM, and 3D reconstruction tasks. It can be noticed that the CNN-DSO performed worse than the original DSO algorithm in sequences 13 and 22 but outperformed the DSO in sequence 39. The LDSO performance was close to the DSO, but it presented a better trajectory in some forward sequences and overcame the DSO in sequence 21. The DSM performed similarly to the rest of the sparse-direct approaches but occasionally presented trajectory loss issues affecting the overall performance. Furthermore, the DynaSLAM considerably outperformed the ORB-SLAM2, especially in challenging sequences like 18, 19, 21, 22, 23, 27, 28, 38, 39, and 40, among others, where the ORB-SLAM commonly failed. However, it occasionally presented trajectory loss and initialization issues. The ORB-SLAM2 optical flow implementation performed slightly worse on forwards and considerably worse on backward, especially in scenes 21, 22, 38, 39, 40, 46, 48, and 50. In contrast, the CNN SVO version considerably reduced the RMSE in most sequences compared to the SVO but still constantly failed in the outdoor sequences 21 and 22, showing random initialization and trajectory loss issues. As reported in [21], the SVO and LSD-SLAM methods had the worst results over the whole dataset, which was why Engel et al. [21] did not include these methods in their study. However, we think it is vital to report such results and the errors attributed to these algorithms' commonly known initialization and trajectory loss errors over the sequences of the TUM-Mono dataset.

The results processed on the TUM-Mono benchmark for the cumulative translation error $e_t$, rotation error $e_r$, scale error $e'_s$, start-segment alignment error $e^s_{align}$, end-segment alignment error $e^e_{align}$, and the translational RMSE $e_{RMSE}$ were gathered in a database defining the method as the categoric variable. The statistical results were processed using R programing language. First, we removed the blank observations for the executions, where each algorithm became lost or could not initialize; so, the Mahalanobis distances were [88]

as a multivariate data cleaning technique to detect and remove the outlier observations. A 22.4577 cut score based on the $\chi^2$ distribution for a 99.999% interval was set up detecting 344 outlier observations ending with a database of 8860 observations.

Then, each dependent variable's normality and homogeneity assumptions were verified to select the appropriate statistical test for comparisons. For example, for the translation error, the *p*-values of $2.2 \times 10^{-16}$ for the DSO, LDSO, CNN-DSO, DSM, DynaSLAM, ORB-SLAM2, DF-ORB-SLAM, CNN-SVO, SVO, and LSD-SLAM methods were obtained in the Lilliefors (Kolmogorov–Smirnov) normality test; so, the sample did not reach the normality assumption. We applied Levene's test obtaining a *p*-value of $2.2 \times 10^{-16}$ for the homogeneity assumption; so, the sample did not meet the homogeneity assumption. The rest of the dependent variables had similar assumptions verification results; thus, it was concluded that the sample was not parametric. Hence, the Kruskal–Wallis test was selected as the general test, with the Wilcoxon signed rank as a pairwise post hoc test. Figure 8 and Table 3 present the results obtained by applying the differences tests.

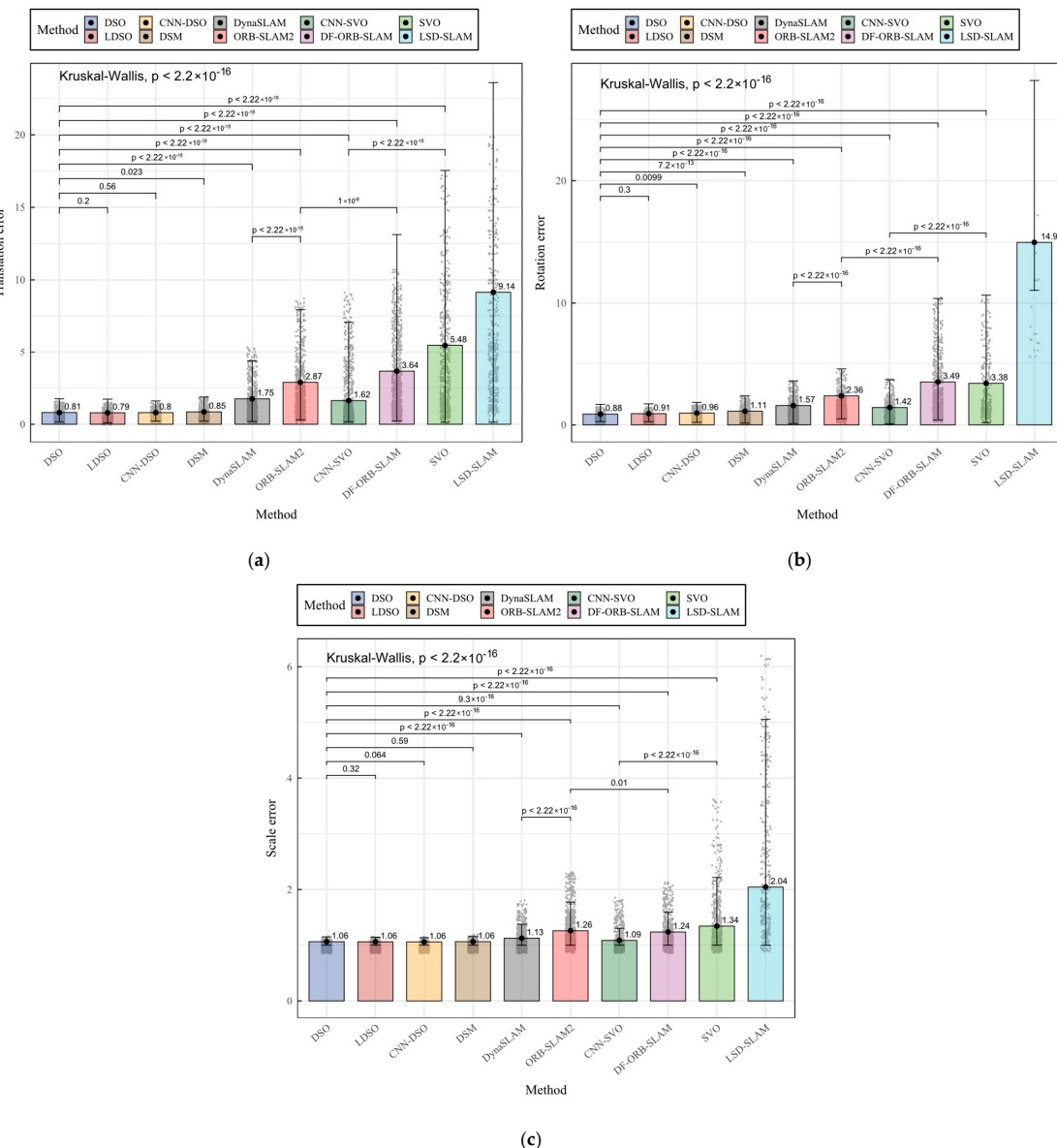

**Figure 8.** Bar plots, box-plot error bars, and Kruskal–Wallis comparisons for the medians of the cumulative errors collected after 1000 runs of each algorithm: the (**a**) translation error, the (**b**) rotation error, and the (**c**) scale error.

**Table 3.** Medians and Kruskal–Wallis comparisons for each algorithm's translation, rotation, and scale errors.

| Method | Translation Error | Rotation Error | Scale Error |
|---|---|---|---|
| Kruskal–Wallis general test | $\chi^2 = 3582.9$ $p_{value} = 2.2 \times 10^{-16}$ | $\chi^2 = 2278.4$ $p_{value} = 2.2 \times 10^{-16}$ | $\chi^2 = 2419.1$ $p_{value} = 2.2 \times 10^{-16}$ |
| DSO | 0.8064585 [a] | 0.8800369 [b] | 1.064086 [ab] |
| LDSO | 0.7892125 [a] | 0.9135608 [ab] | 1.061302 [ab] |
| CNN-DSO | 0.7980411 [a] | 0.9618528 [a] | 1.058849 [a] |
| DSM | 0.8519143 [b] | 1.1117710 [c] | 1.064615 [b] |
| DynaSLAM | 1.7473504 [c] | 1.5730542 [d] | 1.126499 [c] |
| ORB-SLAM2 | 2.8738313 [d] | 2.3585843 [e] | 1.260155 [d] |
| CNN-SVO | 1.6248001 [c] | 1.4159545 [d] | 1.086399 [e] |
| DF-ORB-SLAM | 3.6423921 [e] | 3.4940400 [f] | 1.238232 [f] |
| SVO | 5.4819407 [f] | 3.3772024 [f] | 1.343603 [g] |
| LSD-SLAM | 9.1403348 [g] | 14.9621188 [g] | 2.044298 [h] |

Means with different letters in the same column differ significantly according to the Kruskal–Wallis test and pairwise Wilcoxon signed rank test for $p_{value} \leq 0.05$.

As presented in Figure 8 and Table 3, the sample identified significant differences between the implemented algorithms. By observing the translation error, it can be noticed that the DSO, LDSO, and CNN-DSO methods achieved the most significant performance of the ten evaluated algorithms; despite the DSO performing at 2.18% and 1.05% worse than the DSO and CNN-DSO in this metric, the difference was not significant among them. The DSO, LDSO, and CNN-DSO achieved significantly lower errors than the dense-direct method DSM. The feature-based methods performed significantly worse than the sparse-direct methods, where the DynaSLAM achieved a significantly better performance than the ORB-SLAM2 and DF-ORB-SLAM, reaching a 39.19% and 52.02% translation error reduction, respectively. The CNN-SVO performed slightly worse than the DynaSLAM, but the difference was not significant, while it significantly outperformed its classic version achieving a 47.57% translation error reduction. The LSD-SLAM performed substantially worse in terms of the translation error metric among the ten algorithms.

Regarding the rotation error, the DSO and LDSO achieved significantly better results than the rest of the algorithms. Although the DSO showed an average rotation error reduction close to 3.66%, the difference was not significant. The DSO performed significantly better than its neuronal version in the accumulated rotation error metric. The LDSO performed around 5.02% better than the CNN-DSO, but the difference was not significant. The DSM performed significantly worse than the rest of the sparse-direct methods. The feature-based methods performed worse than the sparse-direct methods in the rotation error metric, where the DynaSLAM achieved a better performance than the ORB-SLAM2 and DF-ORB-SLAM, showing an average error reduction close to 33.30% and 54.97%, respectively. The CNN-SVO performed better than the DF-ORB-SLAM, SVO, and LSD-SLAM in terms of the rotation error metric, significantly outperforming its classic SVO version showing a 58.07% average reduction in the rotation error. The LSD-SLAM performed significantly worse than the other methods in the rotation error metric.

For the scale error metric, the sparse-direct methods, DSO, LDSO, and CNN-DSO, performed significantly better, where the CNN-DSO showed the best performance by an average 0.49% and 0.23% reduction compared to the DSO and LDSO, but the difference was not significant. The DSM performed significantly worse than the CNN-DSO. The feature-based methods performed significantly worse than the sparse-direct methods on the scale error metric, where the DynaSLAM achieved the significantly best performance and an average reduction of 10.60% and 9.02% compared to the ORB-SLAM2 and DF-ORB-SLAM. The CNN-SVO performed significantly better than the feature-based methods, SVO and LSD-SLAM, exhibiting a 19.14% average error reduction compared to its classic version, SVO. Again, the LSD-SLAM performed worst in the scale error metric.

Similarly, the Kruskal–Wallis test was applied as general test, with the Wilcoxon signed rank test, for statistical comparison among the ten methods for the start- and end-segment alignment errors and the overall RMSE. Figure 9 and Table 4 present the results obtained by applying the differences tests.

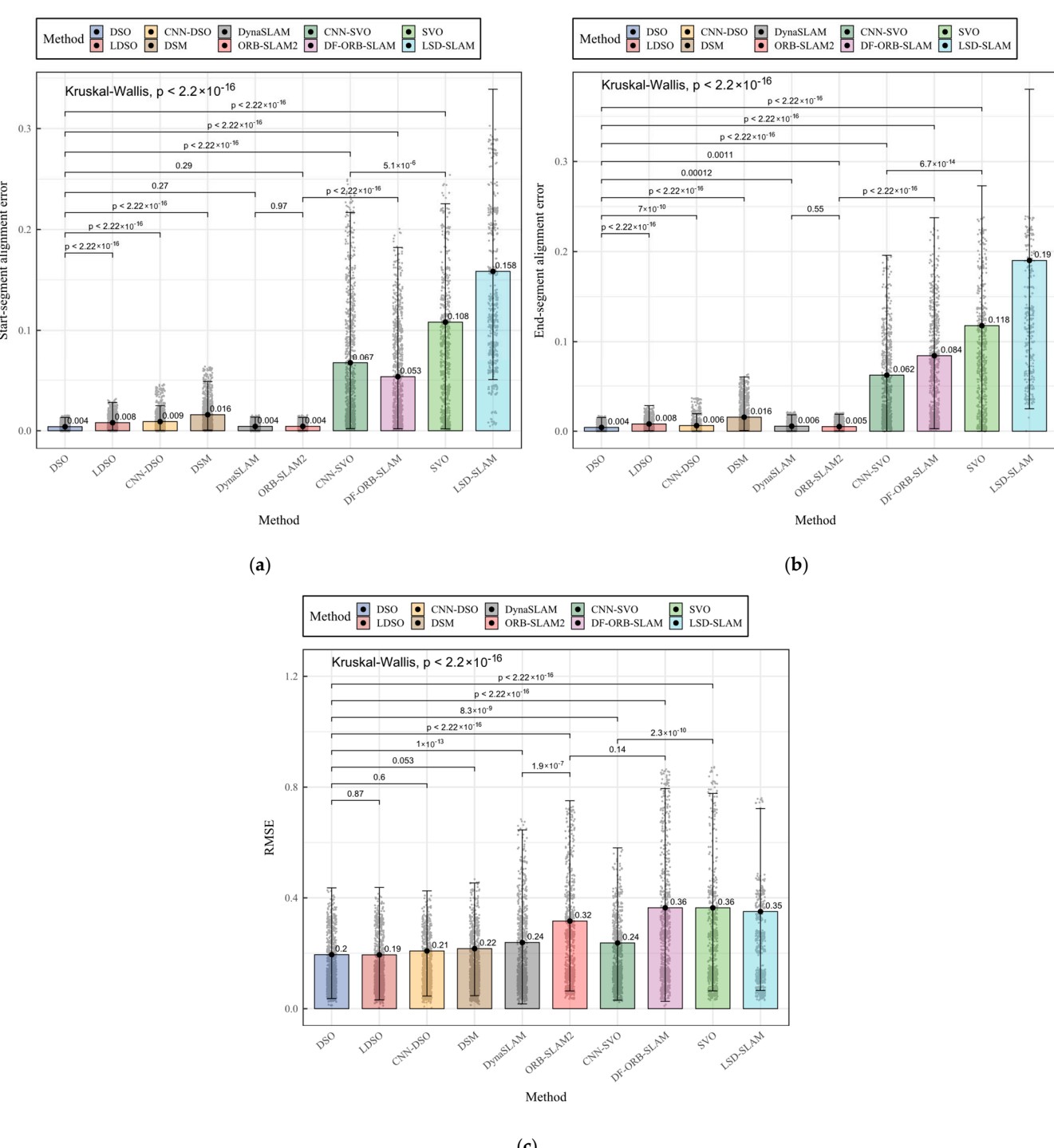

**Figure 9.** Bar plots, box-plot error bars, and Kruskal–Wallis comparisons for the medians of the cumulative errors collected after 1000 runs—(**a**) only the start-segment alignment error, (**b**) only the end-segment alignment error, (**c**) the RMSE for the combined effect on the start and end segments.

**Table 4.** Medians and Kruskal–Wallis comparisons for the translation errors of each algorithm.

| Method | Start-Segment Alignment Error | End-Segment Alignment Error | RMSE |
|---|---|---|---|
| Kruskal–Wallis general test | $\chi^2 = 4575.7$ $p_{value} = 2.2e \times 10^{-16}$ | $\chi^2 = 3718$ $p_{value} = 2.2 \times 10^{-16}$ | $\chi^2 = 530.78$ $p_{value} = 2.2 \times 10^{-16}$ |
| DSO | 0.003974759 [a] | 0.004184367 [a] | 0.1950799 [ab] |
| LDSO | 0.007925665 [b] | 0.008009198 [b] | 0.1944492 [a] |
| CNN-DSO | 0.008987173 [b] | 0.006199582 [c] | 0.2083872 [ab] |
| DSM | 0.015794222 [c] | 0.015537213 [d] | 0.2167750 [b] |
| DynaSLAM | 0.004286919 [a] | 0.005516179 [e] | 0.2389837 [cd] |
| ORB-SLAM2 | 0.004311949 [a] | 0.005102672 [e] | 0.3165024 [e] |
| CNN-SVO | 0.067201999 [d] | 0.062036008 [f] | 0.2373532 [c] |
| DF-ORB-SLAM | 0.053360456 [e] | 0.084420570 [g] | 0.3643844 [e] |
| SVO | 0.108150349 [f] | 0.117753996 [h] | 0.3642558 [e] |
| LSD-SLAM | 0.158469383 [g] | 0.190127787 [i] | 0.3507099 [d] |

Means with different letters in the same column differ significantly according to the Kruskal–Wallis test and pairwise Wilcoxon signed rank test for $p_{value} \leq 0.05$.

Figure 9 and Table 4 show many significant differences between the ten compared methods on the alignment error and RMSE metrics. Regarding the start-segment alignment error, the DSO, DynaSLAM, and ORB-SLAM methods outperformed the rest of the algorithms. Despite the DSO slightly reducing the average start-segment alignment error by around 7.28% and 7.81% compared to the DynaSLAM and ORB-SLAM2, the differences were not significant. The rest of the sparse-direct methods, LDSO, CNN-DSO, and DSM, performed significantly worse than the DSO by an average of 49.84%, 55.77%, and 74.83%, respectively, in the start-segment alignment error metric. For the feature-based methods, the DynaSLAM and DSO performed significantly better than the DF-ORB-SLAM, while the DF-ORB-SLAM achieved an error significantly lower than the CNN-SVO, SVO, and LSD-SLAM. When comparing the CNN-SVO with its predecessor SVO, the difference was significant, where the neural version reduced the start-segment alignment error by an average of close to 37.86%. The LSD-SLAM achieved the worst start-segment alignment error of the ten methods, significantly.

By observing the end-segment alignment error, it was found that the sparse-direct methods significantly outperformed the rest of the compared methods. The DSO significantly outperformed all the evaluated methods, including the rest of the sparse-direct methods, the LDSO, CNN-DSO, and DSM, reducing the average alignment error by around 47.85%, 32.50%, and 73.06%, respectively. In the sparse-indirect category, the DynaSLAM and ORB-SLAM2 performed significantly better than the DF-ORB-SLAM, but even though the ORB-SLAM2 reduced the average end-segment error by approximately 7.49%, the difference was not significant. The CNN-SVO end-segment alignment error was significantly lower than the error of the SVO, reducing this metric by approximately 47.31%. The LDSO performed significantly worse than the rest of the methods.

For the RMSE metric, the sparse-direct methods performed significantly better than the rest, where the LDSO achieved RMSE values around 0.32% and 6.68% lower than the DSO and CNN-DSO; the differences were not significant. The LDSO performed significantly better than the DSM, with an average RMSE around 10% lower. In the feature-based classification, the DynaSLAM performed significantly better than the ORB-SLAM2 and the DF-ORB-SLAM, reducing the RMSE metric by approximately 24.49% and 34.41%, respectively. For the hybrid methods, the CNN-SVO performed significantly better than the SVO, reducing the RMSE by around 34.83%. As with the rest of the metrics, the LSD-SLAM performed significantly worse than the other methods in terms of the RMSE metric.

Finally, in Figure 10, we present the sample trajectories obtained by the three overall best methods evaluated in this comparison study. To exemplify the behavior of the algorithms in different environments, we selected the sequence *seq-02* of the TUM-Mono

dataset as an example for indoors and the sequence *seq-29* as an example for outdoors. In addition, we provide video samples of the execution of each algorithm as supplementary material in the GitHub repository: "https://github.com/erickherreraresearch/MonocularPureVisualSLAMComparison accessed on 16 June 2023", along with all the .txt result files of each algorithm run for reproducibility.

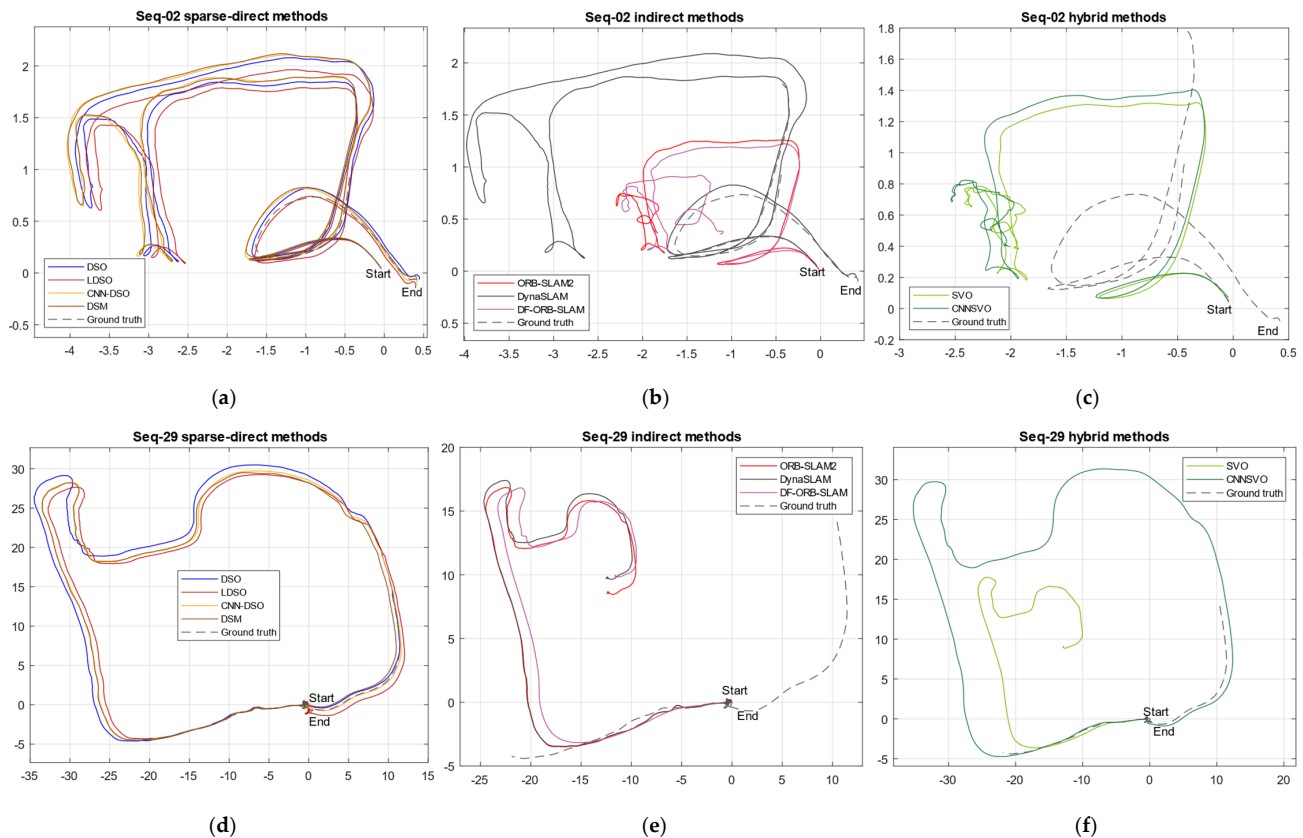

**Figure 10.** Trajectories in the TUM-Mono dataset for the compared sparse-direct (**a**,**d**), indirect (**b**,**e**), and hybrid methods (**c**,**f**). The top row displays the results for the indoor sequence *seq-02*, and the bottom row displays the results for the outdoor sequence *seq-29*. The solid lines represent the trajectory estimated by each algorithm; the dashed lines represent the aligned ground truth.

As depicted in Figure 10, the algorithm's observed behavior ratified this comparative analysis of quantitative results. On the top row, for the indoor sequence, it can be noticed that the sparse-direct methods outperformed the other evaluated methods starting and ending their trajectory pretty close to the ground truth. The indirect methods behaved completely differently, where the system constantly lost the trajectory, accumulating drift and obtaining the wrong scale measures concatenated erroneously when the system achieved relocalization. The DynaSLAM represents an important contribution to the ORB-SLAM2 system because it estimated the trajectory better than the rest, closing the trajectory close to the ground truth, while the rest of the indirect systems lost their trajectories. On the other hand, the hybrid methods performed considerably worse indoors; so, many of the algorithms' runs did not complete the full frame sequence, and the algorithm typically finished its trajectory pretty far from the end-segment ground truth. In the bottom row of Figure 10, it can be noticed again that the sparse-direct methods outperformed the rest of the evaluated systems, with an appropriate bootstrapping in the start segment and a small amount of accumulated drift in the end segment.

In Figure 10e, similar to indoors, the indirect methods suffered from trajectory loss issues despite the fact that the relocalization module typically was able to continue the system execution, accumulating a critical amount of drift during relocalization, making the

estimated trajectory end far from the groundtruth end segment. In the hybrid methods, the SVO suffered from similar issues to the indirect methods. However, it can be noticed that the CNN version of the SVO improved its performance outdoors, differently from indoors, which can be explained because the added CNN MonoDepth module was trained in the Cityscapes dataset, which was mainly trained from outdoor sequences.

## 4. Discussion

In the previous studies of [39,40], the authors compared the DSO, LDSO, ORB-SLAM2, and DynaSLAM algorithms on the same TUM-Mono benchmark, and their findings mostly matched what we observed during this evaluation. However, we extended their study considerably by implementing six additional methods following the taxonomy described in [9] and performed an appropriate statistical analysis to determine the significant differences in each system's performance. Thus, we could observe the classification advantages and limitations for classic geometric-based approaches. The classic approaches can be classified as sparse-indirect, dense-indirect, dense-direct, sparse-direct, and hybrid. For the sparse-indirect, we selected its gold standard, ORB-SLAM2, and we can report that, in our experience, it is an excellent method that demands low computational power and has an average performance not being the best but still working well outdoors. Its main limitations are its poor indoor performance and large drift and scale error accumulation during relocalization. We believe the ORB-SLAM2 has achieved high popularity due to its low computational power consumption and ease of implementation.

For the dense-indirect category, we selected the DF-ORB-SLAM, an ORB-SLAM2 open-source implementation with an additional optical flow estimation module that allows the ORB-SLAM2 to work with more image information. In this case, we observed that the optical flow module increased the computational cost of the algorithm and slightly reduced the occurrence of trajectory loss issues. However, adding image information for feature extraction also introduced noise in the estimation steps, significantly increasing the translation, rotation, scale error metrics, and RMSE. For dense-direct approaches, we selected their gold standard, the LSD-SLAM, one of the first proposed direct methods. In this case, we found that the performance was significantly the poorest of all the evaluated methods. This situation could be due, in particular, to the frequent initialization errors and trajectory loss in most of the benchmark sequences matching what Engel et al. reported in their study [21]. For the sparse-direct classification, we selected the most iconic VO system, the DSO. This system significantly outperformed the methods of the rest of the classic classifications by a large margin, demonstrating an impressive performance indoors and outdoors. The DSO was slightly outperformed by its neuronal version on the scale metric and slightly outperformed by the LSDO in the translation and RMSE metric, but the differences were not significant. This behavior lets us conclude that even its LDSO, CNN-DSO, and DSM extensions do not significantly outperform the DSO, and this method can still be considered a great avenue for future work, improvements, and contributions.

Additionally, it must be mentioned that, in contrast to what was reported in [31], the DSM method did not outperform its predecessor, the DSO, in the TUM-Mono dataset, even after implementing a complete SLAM pipeline to extend the DSO. We believe the DSM is a robust system that can contribute to the sparse-direct category. However, the authors used the extended pinhole radial-tangential camera model instead of the complete photometrical camera calibration provided in the original DSO, including the intrinsic, photometric, and nonparametric vignette calibration. As reported in [40], this situation considerably contributed to the correct execution of the algorithm and allows it to obtain the best results. Then, for the hybrid approaches, we tested the SVO, which combines direct and indirect formulation paradigms in its pipeline. The SVO was second to last in our comparison, being significantly outperformed by most of the methods in this study, only performing significantly better than the LSD-SLAM, in line with what was reported in [21]. However, with this analysis, we could observe that it at least performed better than the dense-direct method. The SVO is also a popular method in the robotics research field.

We believe that this is caused by its early appearance in 2014, low computational power requirement, and open-source availability for implementation with C++ or ROS.

For machine learning classification, there are many ML implementations available in the literature [2,10,14,20,32–36,65,66,89–95], and many of them include open source code implementations [2,17,19,42,44–53,96]. Nevertheless, most methods were formulated to work with more than one input mode, like RBD-D or INS, and their code implementations did not include monocular running instructions or did not include their monocular RGB pipeline, e.g., [2,42,53]. Other open-source code implementations required additional external information for running, like optical flow or feature extractors, that were not included as open source, e.g., [50]. For such reasons, we selected three ML versions of the classic approaches of the DSO, ORB-SLAM2, and SVO: CNN-DSO, DynaSLAM, and CNN-SVO. Therefore, we concluded that the CNN-SVO significantly outperformed its predecessor in all the metrics, the DynaSLAM significantly outperformed the ORB-SLAM2 in all the metrics except for the end-segment alignment error where the difference was not significant, and the CNN-DSO significantly outperformed its classic version only in the rotation error metric. Here, we can mention that in contrast to what is reported in the CNN-DSO official repository [15], where the algorithm was evaluated in the first eleven sequences of the KITTI dataset, after testing the algorithm in a larger dataset indoors, outdoors, and a large variety of motion patterns, the CNN-DSO only slightly outperformed the DSO in the scale error metric, but the observed difference was not significant, while the DSO still outperformed it in rotation and in the start- and end-segment alignment error metrics. In addition, during the evaluation, it was observed that the algorithm introduced a considerable number of outlier points in the obtained 3D reconstruction. Thus, we can point out that machine learning studies are making vital contributions to enhancing the monocular VO, SLAM, and 3D reconstruction systems. Table 5 summarizes the observed advantages and limitations of the evaluated methods based on the experience of implementing and running the algorithms.

**Table 5.** Practical advantages and limitations of the evaluated methods.

| Method | Category | Advantages | Limitations |
|---|---|---|---|
| ORB-SLAM2 [54] | Classic sparse-indirect | Low computational cost. Multiple input modes. Ease of implementation. Robustness to multiple environments. | Trajectory loss issues. Accumulation of drift while relocalizing. Sparse 3D reconstruction. |
| DF-ORB-SLAM [16] | Classic dense- indirect | Low computational cost. Reduction in trajectory loss issues. | Introduction of noise for trajectory estimation. Accumulation of drift on relocalization. Sparse 3D reconstruction. Significant reduction in the performance of ORB-SLAM2. |
| LSD-SLAM [29] | Classic dense-direct | Low computational cost. More detailed 3D reconstruction, but with the presence of outliers. More information in the final 3D reconstruction. | Poorest performance of the evaluated methods. Initialization issues. Trajectory loss issues. |
| DSO [21] | Classic sparse-direct | Low computational cost. Ease of implementation. More detailed and precise 3D reconstruction. Robust to multiple environments and motion patterns. Best performance of all methods in most of the metrics. | Requirement for a specific complex camera calibration. Slightly, but not significantly, lower performance than the LDSO in the translation and RMSE metrics. |

**Table 5.** *Cont.*

| Method | Category | Advantages | Limitations |
|---|---|---|---|
| SVO [13] | Classic hybrid | Low computational cost. Good documentation and open-source availability for implementation in diverse configurations. | Frequent trajectory loss issues. Initialization issues. Critical execution errors due to the absence of a relocalization module. |
| LDSO [30] | Classic sparse-direct | Low computational cost. Similar to DSO, detailed and precise 3D reconstruction. Ease of implementation. Loop closure module. Slightly but not significantly better performance than the DSO in translation and rotation error. Best performance in the translation and RMSE metrics (compared to DSO), but without considerable difference. | Requirement of a specific complex camera calibration. Significantly worse performance than the DSO in the end-segment error metric. |
| DSM [31] | Classic sparse-direct | Detailed and precise 3D reconstruction. Robust execution in most of the environments and motion patterns. Complete and interactive GUI. | Requirement of more computational capabilities than the rest of the sparse-direct methods. Significant underperformance compared to most of the sparse-direct methods. |
| CNN-DSO [15] | ML sparse-direct | Detailed and precise 3D reconstruction. Robust to multiple environments and motion patterns. Best performance in scale error metric. | Presence of outliers in the 3D reconstruction. Significantly better performance in the rotation error metric by the DSO. Difficult to implement. Specific hardware requirement. |
| DynaSLAM [32] | ML sparse-indirect | Multiple input modes. Ease of implementation. Robustness to multiple environments. Ability to detect, segment, and remove information of moving objects. Especially recommended for dynamic environments. Fewer trajectory loss issues than ORB-SLAM2. | Accumulation of drift while relocalizing. Sparse 3D reconstruction. Increase in complexity over the ORB-SLAM2. Specific hardware requirement. |
| CNN-SVO [11] | ML hybrid | Considerable reduction in the trajectory loss issues compared to the SVO. Initialization issues. Reduction in the number of execution issues compared to the SVO. Improved performance over the ORB-SLAM2 in the rotation, translation, scale, and RMSE metrics. Significant improvement over its classic version in all the metrics. | Considerable presence of outliers in the 3D reconstruction. Imprecise and sparser 3D reconstruction. Complex implementation. Specific hardware requirement. |

## 5. Conclusions

In this article, the most representative open source monocular RGB SLAM and VO available implementations were tested following a taxonomy to determine the advantages and limitations of each method and classification, providing the reader a guide to correctly select the method that fits their needs or to select a path to make future contributions to the tested methods and classifications. After experimentation, it can be concluded that the monocular SLAM and VO methods need to be evaluated on larger datasets in a large variety of environments, motion patterns, and illumination conditions to be effectively compared with the state of the art, as demonstrated in this study for methods like the DSM, CNN-DSO, and DF-ORB-SLAM that did not match the expected results on the TUM-Mono dataset.

The sparse-direct category of the taxonomy achieved the significantly best results among all the ten methods in the translation, rotation, scale, and RMSE metrics outputting the most detailed and precise 3D reconstructions of the tested methods. At second best, the sparse-indirect category achieved good ego-motion estimation but output sparser 3D reconstructions that might not be suitable for many applications, presenting trajectory loss issues and evidencing worse performance indoors. Additionally, by including three machine learning-based methods and comparing them with their classic versions, we can conclude that the integration of machine learning significantly improves the performance of the SLAM or VO systems and should be considered as a future research direction to overcome the limitations of each system. Integrating CNN information for the estimation steps contributes to mitigating monocular systems' commonly known scale ambiguity issue. This behavior was demonstrated in each ML method's significant scale error reduction compared to their classic versions.

Through experimentation, refs. [6,40] concluded that the great majority of the alignment error originated in the accumulated drift, independent from the noise in the ground truth that can be registered with any SLAM or VO, which allows using all the metrics using the ground truth of only the start and end segments. We agree and confirm that this conclusion allowed us to compare a wide variety of methods coming from different configurations and classifications that output trajectories in different scales and orientations, which can be efficiently compared after the proposed benchmark alignment method.

**Author Contributions:** Investigation, E.P.H.-G.; conceptualization, E.P.H.-G. and D.H.P.-O.; Methodology, E.P.H.-G. and J.C.T.-C.; supervision, A.R., J.C.T.-C. and D.H.P.-O.; writing—original draft preparation, E.P.H.-G.; writing—review and editing, E.P.H.-G., J.C.T.-C., D.H.P.-O. and A.R.; funding acquisition, D.H.P.-O. All authors have read and agreed to the published version of the manuscript.

**Funding:** This work was supported by the SDAS Research Group (www.sdas-group.com accessed on 16 June 2023).

**Informed Consent Statement:** Not applicable.

**Data Availability Statement:** Video samples of each algorithm's indoor and outdoor executions are provided as Supplementary material in the GitHub repository: https://github.com/erickherrerares earch/MonocularPureVisualSLAMComparison along with all the .txt result files of each algorithm run for reproducibility.

**Conflicts of Interest:** The authors declare no conflict of interest.

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
