# Peer review of "A Comparison of Monocular Visual SLAM and Visual Odometry Methods Applied to 3D Reconstruction"

_applsci, doi:10.3390/app13158837_

Round 1

Reviewer 1 Report

The article examines various approaches to monocular 3D reconstruction, such as SLAM, VO, and SFM, employing a classification system to assess their effectiveness. The findings indicate that sparse-direct techniques surpassed alternative methods while incorporating machine learning approaches notably enhanced the performance of geometric-based methods. 

I want to receive an answer to the following:

It is important to note that the evaluation was conducted using a particular dataset, potentially limiting its generalizability to diverse scenarios. 

Furthermore, the paper did not provide a comprehensive examination of the computational expenses associated with each approach, which holds significance for real-world applications.

Reviewer 2 Report

A comprehensive and meaningful comparison study for VSLAM/VO/SFM applications. Some of my suggestions for the article are as follows:

1) In the introduction, there should be a paragraph describing SLAM, VO and SFM. For this, citations should be made. For example, the following study can be used.

https://doi.org/10.1007/978-3-030-75472-3_7

2) There is a lot of general information for section 2.1. However, there is no reference. This information should be verified with references. For example, the following study can be used.

  https://doi.org/10.1155/2021/2054828

3) This study compares different types of sparse, indirect, dense, direct approaches suggested in VSLAM and VO studies in general. However, the related works section is very inadequate for this comparison. Numerous methods should be introduced and criticized in this section. New methods such as HVIONet, VIIONet, StereoVO should be mentioned.

4) Why is monocular 3D reconstruction an ill-posed problem?

5) A comparison was also made over the datasets. Therefore, different datasets should also be mentioned in this article. For example YTU,WHUVID, VOID, etc.

6) In line 483, the equal numbers are incorrect. Also, this is a comparison study. But there are many equations. The equation of the existing methods can be explained in a simpler way. Is Equation 12 - Equation 17 necessary or why is it important?

7)"Finally, in Figure 10, we present son sample trajectories obtained by the three overall best methods evaluated in this comparison study." What does "son" mean?

8) There are many errors in the use of abbreviations in the article. It should be corrected. For example, ML.

Round 2

Reviewer 2 Report

The authors revised the article in accordance with the comments. The article is acceptable.